# Can Optimization Trajectories Explain Multi-Task Transfer?

**David Mueller**                                                                      *dam@jhu.edu*
*Department of Computer Science*
*Johns Hopkins University*

**Mark Dredze**                                                                 *mdredze@cs.jhu.edu*
*Department of Computer Science*
*Johns Hopkins University*

**Nicholas Andrews**                                                               *noa@cs.jhu.edu*
*Department of Computer Science*
*Johns Hopkins University*

**Reviewed on OpenReview:** *https://openreview.net/forum?id=QQE5j2OsLW*

## Abstract

Despite the widespread adoption of multi-task training in deep learning, little is understood about how multi-task learning (MTL) affects generalization. Prior work has conjectured that the negative effects of MTL are due to optimization challenges that arise during training, and many optimization methods have been proposed to improve multi-task performance. However, recent work has shown that these methods fail to consistently improve multi-task generalization. In this work, we seek to improve our understanding of these failures by empirically studying how MTL impacts the optimization of tasks, and whether this impact can explain the effects of MTL on generalization. We show that MTL results in a **generalization gap**—a gap in generalization at comparable training loss—between single-task and multi-task trajectories *early into training*. However, we find that factors of the optimization trajectory previously proposed to explain generalization gaps in single-task settings cannot explain the generalization gaps between single-task and multi-task models. Moreover, we show that the amount of gradient conflict between tasks is correlated with negative effects to task optimization, but is not predictive of generalization. Our work sheds light on the underlying causes for failures in MTL and, importantly, raises questions about the role of general purpose multi-task optimization algorithms. We release code for all of our experiments and analysis here: https://github.com/davidandym/Multi-Task-Optimization

## 1 Introduction

Multi-task learning (MTL)—the simultaneous optimization of multiple related tasks—has a long history in machine learning (Caruana, 1993). By learning from additional related signals during training, multi-task learning can yield models with stronger generalization than single-task models; however, these additional training signals may not always benefit one another, and MTL can also lead to models which generalize *worse* than single-task models (Figure 1; Zhang et al., 2023). Prior work has conjectured that the negative impacts of multi-task training on task generalization occur due to *optimization challenges* that arise during joint training of multiple objectives simultaneously. Consequently, a number of specialized multi-task optimizers (SMTOs; Kurin et al., 2022) have been proposed to address these optimization challenges in order to improve the generalization of multi-task models (Chen et al., 2018; Sener & Koltun, 2018; Yu et al., 2020,

*inter alia*). However, recently Kurin et al. (2022) and Xin et al. (2022) found that these SMTOs actually fail to consistently improve the performance of MTL models over the baseline of uniformly aggregated SGD.

SMTOs are developed on the hypothesis that large differences between the gradients of tasks (often termed gradient conflict) gives rise to certain optimization challenges that naive SGD will not overcome. As a result, many SMTOs are motivated by proving convergence in simplified settings (e.g. Yu et al., 2020; Chen et al., 2020) or demonstrating the method's superiority on toy optimization problems (e.g. Liu et al., 2021a) where such optimization challenges exist. However, in deep learning, it is not clear that these optimization challenges, or lack thereof, can explain the mechanisms that drive *transfer* (the impact of MTL on generalization). For instance, in single-task learning, large-batch training may result in worse generalization than small-batch training despite large-batch training leading to better training loss optimization (Smith et al., 2018). The disconnect between our understanding of how MTL impacts optimization and how MTL impacts generalization may explain the recent claims that SMTOs frequently do *not* improve MTL performance (Kurin et al., 2022; Xin et al., 2022). In this work, we aim to bridge this disconnect by asking how the **impact of MTL on a task's optimization can explain its effect on that task's generalization**.

We approach this question first by comparing multi-task and single-task training loss trajectories to one another, studying the *trade-off* between tasks—both with respect to optimization and generalization—within an MTL setting, to understand why MTL benefits some tasks while simultaneously hurting others. Then, by studying the trajectories of a few target-tasks as the set of auxiliary tasks changes, we study how the *amount* of task conflict in MTL impacts task optimization trajectories, and whether this impact is predictive of transfer. Specifically, our research questions and contributions are:

- **What can training loss minimization tell us about multi-task transfer?** In §3 we compare multi-task and single-task generalization by the total task training loss at each epoch across 5 multi-task settings. We find that transfer (both positive & negative) is observable across comparable training losses as early as a few epochs into training and is often maintained as a generalization gap between comparable training loss throughout the rest of training (e.g. Figure 1).

- **Can transfer be explained by factors of the training trajectory beyond training loss?** In §4 we study whether certain factors of optimization trajectories, previously connected to generalization in deep learning, are correlated with the impact of MTL on generalization. We find that, while MTL does impact these factors (gradient coherence, early-stage Fisher information, and loss surface sharpness), the effect of MTL on task performance is not explained by these factors. Moreover, we show that, when SMTOs impact task performance, their effect is also not explained by their impact to these factors.

- **How does the amount of conflict impact optimization and generalization?** Finally, in §5 we study how varying the gradient conflict that a task experiences impacts the factors that we study above. We find that a high amount of gradient conflict is correlated with negative impacts to all of the factors we study in §4. However, we simultaneously find that the amount of conflict has little-to-no correlation with task generalization and the effect of gradient conflict on optimization does not predict the benefit of different auxiliary tasks.

Our findings demonstrate a (current) inability of optimization trajectories—and the impact of MTL on them—to explain multi-task transfer. Importantly, our results make clear that (a) our current understanding of optimization and generalization in deep learning is not capable of predicting transfer from MTL and, as a result, (b) it is not clear what optimization challenges must be overcome in MTL, or what general purpose optimization algorithms need to tackle, to broadly improve the performance of multi-task models.

## 2 Background and Preliminaries

### 2.1 Multi-Task Optimization and Transfer

In a canonical multi-task setting, there are $K$ tasks, where each task, $k$, consists of a dataset, $S_k = \{(x_i^{(k)}, y_i^{(k)})\}_{i=1}^{N_k}$, drawn from some task distribution, $S_k \sim \mathcal{D}_k$. [1] Given a network $f$ with parameters

---

[1]In this work we will focus on the settings where we assume that $S_k$ is drawn i.i.d. from $\mathcal{D}_k$ and in which we are interested in generalizing over $\mathcal{D}_k$. This is the setting where transfer and MTL optimization is most commonly studied.

$\Theta = \{\theta, \phi_{k_1}, \dots, \phi_{k_K}\}$, where $\theta$ are shared across all tasks and $\phi_k$ are specific to task $k$, our goal is to solve the following minimization problem:

$$\min_{\Theta} \left\{ \mathcal{L}^{MT}(\Theta) = \sum_{k \in K} w_k \mathcal{L}_k(\theta, \phi_k) \,\middle|\, \mathcal{L}_k(\theta, \phi_k) = \mathbb{E}_{(x,y) \in S_k} \ell_k(f_{\theta, \phi_k}(x), y) \right\} \tag{1}$$

where $\ell_k$ is some (potentially task-specific) loss function and $w_k$ are task weights that are typically set to $w_k = 1$ to reflect *a priori* no preference on distinct task objectives. The hope of multi-task learning is that solving equation 1 will yield a solution $\Theta^*_{MT}$ that results in better generalization for each task than the solution found by solving each task individually $(\theta^*_{ST}, \phi^*_{ST})$. More formally, let $\mathcal{E}_k(\theta, \phi_k) = \mathbb{E}_{(x,y) \sim \mathcal{D}_k} a(f_{\theta, \phi_k}(x), y)$ be a measure of generalization on unseen samples of the task distribution $\mathcal{D}_k$ for task $k$, given some metric $a$; our hope is that multi-task *transfer* is positive, which occurs when $\mathcal{E}_k(\theta^*_{MT}, \phi^*_{MT}) - \mathcal{E}_k(\theta^*_{ST}, \phi^*_{ST}) > 0$.

A naive solution to optimizing Equation 1 is to leverage the uniform multi-task gradient (UMTG) for $\theta$:

$$\nabla_\theta^{MT}(\Theta, B) = \frac{1}{C} \sum_{k \in K} w_k \nabla_\theta \mathcal{L}_k^B(\theta, \phi_k) \; ; \; w_k = 1 \, \forall k \in K \tag{2}$$

where $\mathcal{L}_k^B$ is the loss of task $k$ over a randomly sampled batch $B$, and $C$ is the *scaling factor*.[2] The UMTG is thought to frequently result in poor performance for multi-task learning, often yielding *worse* generalization than single-task learning, a phenomenon named **negative transfer** (Zhang et al., 2023). Negative transfer is often attributed to **gradient conflict** in the parameters of $\theta$, which is typically thought to arise in two different manners (Liu et al., 2021b; Javaloy & Valera, 2022): directional conflict occurs when the angle between task gradients is high, preventing any single direction from locally optimizing all tasks jointly (e.g. Yu et al., 2020; Wang et al., 2021); separately, magnitude conflict can arise when the magnitude of task gradients are disparate, resulting in the under-optimization of certain tasks (e.g. Chen et al., 2018).

To improve the performance of multi-task learning, prior work has focused on developing specialized multi-task optimizers (SMTOs, Kurin et al., 2022) which aim to improve multi-task optimization by mitigating task conflict during training. Although these methods hope to improve the *generalization* of multi-task models, they directly target ways to improve *minimization of* Equation 1 by addressing conflict between tasks during training (Navon et al., 2022; Liu et al., 2021a; Chen et al., 2020; Javaloy & Valera, 2022, *inter alia*).[3] However, in deep learning better training loss minimization does not always lead to better generalization (e.g. Smith et al., 2020) and indeed Kurin et al. (2022) and Xin et al. (2022) recently demonstrated that SMTOs often do *not* improve multi-task performance over the UMTG. In this work we empirically study *how* gradient conflict impacts task optimization, and whether this impact can explain the effect of multi-task learning on task *generalization*.

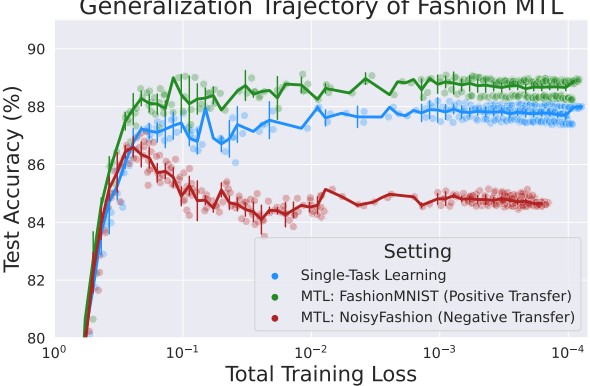

Figure 1: Fashion1 (§2.2) training loss by generalization for the single-task setting (blue curve) and two multi-task settings (red and green curves). The impact of multi-task training on test accuracy (positive *and* negative) is detectable early into the training trajectory, at comparatively high training losses.

## 2.2 Experimental Setup

Following work on multi-task learning and gradient conflict, we consider MTL architectures which leverage the *shared-encoder* architecture, i.e. $f_{\theta, \phi_k} = h_{\phi_k} \circ g_\theta$ where $g_\theta$ is an encoder model which maps inputs into

---

[2]$C$ is typically set to $|K|$, yielding a uniform average gradient, or 1, yielding a uniformly summed gradient. Unless otherwise noted, we set $C = 1$ in our experiments, following the best practice suggested by Mueller et al. (2022).

[3]For example, methods like PCGrad (Yu et al., 2020; Wang et al., 2021) are motivated by aligning task gradient directions such that convergence on all tasks is faster and avoids local minima.

a representation space shared across all tasks and $h_{\phi_k}$ is a task-specific head that maps representations to task-specific predictions. We consider the following MTL settings:

- **FashionMTL:** FashionMTL is a synthetic MTL setting that we construct from the FashionMNIST task (Xiao et al., 2017) in which we have a target-task, an ideal auxiliary task that yields positive transfer, and an uninformative auxiliary task that yields negative transfer. We split the original FashionMTL task in two (with equal class balance in both halves) and treat each half as a separate task (Fashion1 and Fashion2), creating two tasks of size $25,000$ samples each. Additionally, we create a third task, which we call "NoisyFashion," in which we randomly permute the labels of the Fashion2 data. We expect the Fashion1 data to observe positive transfer when trained with the Fashion2 task, and negative transfer when trained with the NoisyFashion task.
- **MNISTS**: The MNISTS multi-task setting (Hsieh & Chen, 2018) consists of 3 MNIST-like tasks: MNIST (LeCun et al., 1998), a 10-class digit classification task; FashionMNIST (Xiao et al., 2017), a 10-class clothing classification task; and NotMNIST, a 26-class English letter classification task. All tasks contain $50,000$ training samples and $10,000$ test samples and all inputs are $28 \times 28$ greyscale images.
- **CIFAR-100:** CIFAR-100 (Krizhevsky, 2012) is a hierarchical 100-class image classification dataset; these class hierarchies can be separated into 20 individual 5-class classification tasks, e.g. Household Electronics classification or Aquatic Mammals classification, each consisting of around $2,500$ samples.
- **CelebA:** CelebA (Liu et al., 2015) is an image classification dataset consisting of celebrity images; each of the $160,000$ images is labeled with 40 binary attributes, which each constitute a classification task.
- **Cityscapes:** The Cityscapes (Cordts et al., 2016) dataset consists of $60,000$ images of urban streets and we follow the setup of Sener & Koltun (2018) and cast it as an image segmentation problem with two tasks: per-pixel 7-class semantic segmentation and pixel-wise depth estimation.
- **GLUE:** The GLUE dataset (Wang et al., 2018) is a benchmark of 8 NLP tasks. 7 tasks are classification tasks, ranging from Natural Language Inference to Grammatical Correctness, and one task is a regression task (Semantic Similarity). The amount of data per task can vary significantly.

For every training trajectory we study, we consider 3 random seeds after selecting hyper-parameters based on the best validation performance out of an initial hyper-parameter sweep. To maintain comparability of individual task trajectories, single-task and multi-task models within a single MTL setting are trained for the same number of steps, with the same optimizer and $C$ (the scaling factor) set equal to $1$.[4]

## 3    What Does the Training Loss Trajectory Tell Us About Transfer?

Although many multi-task optimization methods operate on the theoretical assumption that achieving lower training loss will lead to improved generalization—performance on held-out data from the training distribution—it is well established that, in deep learning, distinct trajectories can generalize very differently at *identical* training loss (Hochreiter & Schmidhuber, 1997; Jastrzębski et al., 2018; Huang et al., 2020). For instance, due to underspecification, different solutions may latch onto different features of the data, leading to overfitting or poor robustness despite achieving low training loss (D'Amour et al., 2022). Moreover, a significant amount of prior work has posited that many properties of the final solution of a training run, including generalization, are determined early into training (Leclerc & Madry, 2020; Jastrzebski et al., 2020; Frankle et al., 2020a; Fort et al., 2020; Frankle et al., 2020b; Juneja et al., 2023). However, it is not known whether, in practical deep learning settings, the failures of multi-task training are due to how gradient conflict affects *convergence* on task training loss (i.e. how the loss is optimized towards the end of training) or how gradient conflict alters properties of optimization throughout training.

### 3.1    Multi-Task Transfer Occurs Early Into Training

We begin by empirically demonstrating that the value of the training loss near convergence is incapable of explaining positive and negative transfer in the 5 MTL settings we consider. For each setting in §2.2 we compare the generalization trajectories of single-task and multi-task training as they pass through regions of

---

[4]See Appendix B for more details around models, datasets, and hyperparameters.

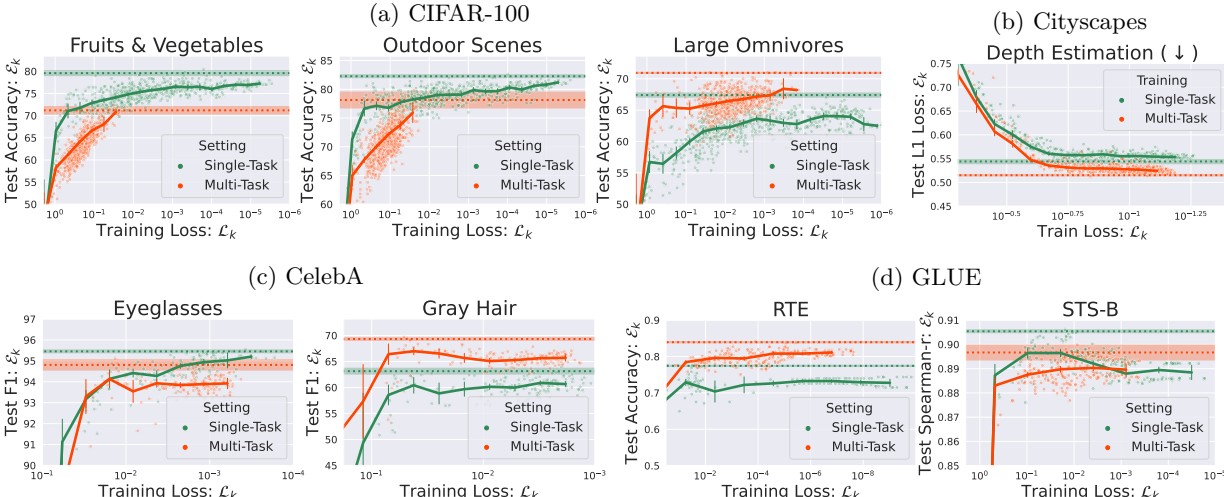

Figure 2: Generalization ($\mathcal{E}_k$) versus Loss ($\mathcal{L}_k$) curves for tasks which exhibit positive or negative multi-task transfer in 4 multi-task settings (for more tasks, see Appendix E). In general, multi-task trajectories converge to a higher training loss than single-task trajectories, meaning gradient conflict stops optimization early. However, *transfer* (positive and negative) is exhibited as a generalization gap between single-task and multi-task trajectories at comparably high training losses, i.e. transfer can be observed early into training. In other words, **multi-task transfer is a property of how gradient conflict impacts the early phase of learning**, rather than a property of how well the task training loss is minimized. Therefore, negative transfer must be explained by higher order factors of the optimization trajectory than the training loss.

similar training loss. More specifically, we evaluate the full training loss ($\mathcal{L}_k$) and generalization ($\mathcal{E}_k$) for at every epoch and we plot generalization by training loss, allowing us to study how MTL generalization differs from STL generalization across *comparable training loss* throughout training. In Figure 1 we plot these trajectories for single-task and multi-task learning in the FashionMTL setting. We see that multi-task training with additional FashionMNIST data leads to positive transfer, whereas training with noisy FashionMNIST data leads to negative transfer. However, in both cases, this transfer occurs as a **generalization gap**—a difference in generalization between two trajectories at comparable training loss—early into training and is then maintained throughout the rest of optimization.[5] In this FashionMTL setting, this gap is observable in trajectories at a training loss over 3 orders of magnitude higher than the eventual training loss at convergence ($\sim 10^{-4}$). This result is surprising if we hold the assumption that negative transfer in MTL arises because gradient-conflict *stops learning early*; however, it matches the intuition put forth by prior work on single-task learning suggesting that certain properties of a training trajectory are determined very early into training (e.g. Leclerc & Madry, 2020; Frankle et al., 2020b, etc.).

In Figure 2, we compare generalization versus loss curves for tasks in 4 additional Multi-Task settings (CIFAR-100, Cityscapes, CelebA, & GLUE), focusing on tasks which either benefit or suffer significantly from MTL in terms of their final model accuracy.[6] Across all settings, we see that tasks which exhibit high amounts of positive or negative transfer also incur generalization gaps between STL and MTL trajectories comparatively early into training, well before either trajectory converges. These generalization gaps are exhibited as significant differences (outside of 2 standard deviations, demonstrated by error bars) between multi-task and single-task trajectories for loss values as early as $10^0$ or $10^{-1}$, in some cases over 6 orders of magnitude higher than the eventual training loss. Importantly, while prior work on multi-task learning has operated on the assumption that the key to improving multi-task performance is tackling challenges in

---

[5]We adopt the term "generalization gap" from literature surrounding the large-batch generalization gap (Keskar et al., 2017), the phenomenon where large-batch models generalize worse than small-batch models across identical training losses.

[6]For more complete task comparisons, see Appendix E. Note that not all tasks see significant transfer (either positive or negative) from multi-task training; we are primarily interested in those tasks which do see significant changes to generalization.

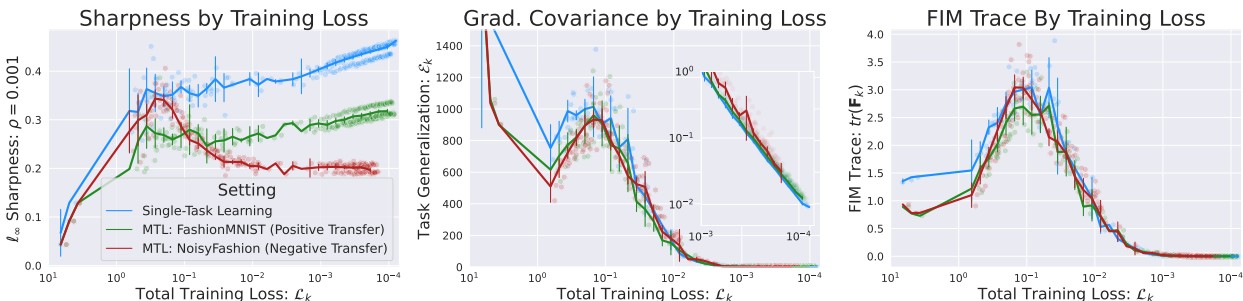

Figure 3: **Factors of the optimization trajectory are unable to simultaneously explain negative and positive transfer**. We plot the trajectories of factors of the loss surface (sharpness, gradient covariance, and Fisher information) for FashionMTL, corresponding to the generalization trajectories in Figure 1 (similar plots for the other multi-task settings are shown in Appendix E). We expect to see the red trajectory, which yields negative transfer, exhibit worse optimization properties than the single-task trajectory (blue curve) and vice-versa for the green curve (positive transfer). Regardless of whether multi-task training resulted in negative or positive transfer, multi-task trajectories (green and red curves) exhibit better optimization properties (e.g. lower sharpness or early-phase FIM "explosions") than single-task trajectories (blue curve).

minimizing the training loss, here we see instead that multi-task performance is driven by factors that are implicit to the zeroth order training loss (the value of $\mathcal{L}_k$) and are determined early into training.

> When multi-task training has a significant impact on task generalization, this impact arises as a **generalization gap** between single-task and multi-task trajectories *early into training*. In other words, transfer must be explained by factors of optimization that go beyond the training loss minimization.

## 4 Can Factors of the Optimization Trajectory Explain Transfer?

In §3 we find that tasks which experience positive or negative transfer from MTL exhibit transfer as a generalization gap early into training. Thus, any theory of (and subsequent method to improve) the trade-off between tasks in a multi-task problem must explain how MTL impacts factors of optimization which are implicit to the zeroth order training loss but connected to generalization. Prior work on generalization in single-task models has proposed several factors of the training trajectory to explain generalization gaps between models (e.g., surface sharpness (Hochreiter & Schmidhuber, 1997)). Here we ask whether certain factors of the loss surface along the optimization trajectory—specifically, sharpness, Fisher information, and gradient coherence—are capable of explaining positive and negative transfer in multi-task settings.

**Loss Surface Sharpness:** The *sharpness* of the region around where a solution lies has long been associated with its generalization in deep learning, both empirically and theoretically (Hochreiter & Schmidhuber, 1997; Keskar et al., 2017; Dziugaite & Roy, 2017; Huang et al., 2020). We follow Andriushchenko et al. (2023) and adopt the elementwise-adaptive worst-case-$\ell_\infty$ $|B|$-sharpness measure:

$$\text{Sharpness}(\theta) = \mathbb{E}_{B \sim S_k} \max_{||\,|\theta|^{-1}\epsilon||_\infty < \rho} \mathcal{L}_k^B(\theta + \epsilon) - \mathcal{L}_k^B(\theta) \tag{3}$$

where $\rho$ is an upper bound on the adaptive $\ell_\infty$ norm of the perturbation, $|B|$ is batch-size, and $S_k$ is the training dataset. Intuitively, this metric measures the maximum change in loss in a radius around the solution $\theta$. We set $|B| = 128$ and $\rho = 10^{-3}$, and we truncate $S_k$ to be of size 2048.

**Gradient Coherence:** To explain the large-batch generalization gap, Smith et al. (2021) leverage backwards analysis to derive an implicit bias term which biases optimization towards regions of the loss surface where the *gradient covariance* is low (i.e. where gradients are "coherent"; Chatterjee & Zielinski, 2022). More specifically, Smith et al. (2021) show that large learning rate and small-batch training may implicitly optimize

| Dataset | Task | $\Delta\mathcal{E}_k$ (Transfer) | Factors of Optimization | | |
| | | | $\Delta$ Sharpness | $\Delta$ FIM Trace | $\Delta$ Coherence |
|---|---|---|---|---|---|
| CIFAR-100 | Fruits And Vegetables | -8.81 | 0.17 | 8.11 | 789.95 |
| | Large Carnivores | -3.39 | 0.04 | -212.92 | -0.26 |
| | Outdoor Scenes | -5.11 | 0.13 | -45.61 | 697.43 |
| | Medium Mammals | -2.89 | 0.05 | 30.69 | 10.85 |
| | Fish | 3.49 | 0.05 | 19.33 | 10.79 |
| | Large Herbivores | 3.92 | 0.05 | 49.69 | 11.06 |
| | Invertebrates | 3.53 | 0.07 | 19.80 | 1.42 |
| CelebA | Blurry | -2.73 | 0.31 | 2.20 | 90.39 |
| | Eyeglasses | -1.20 | 0.32 | 0.31 | 7.90 |
| | Double Chin | 5.30 | 0.33 | 0.51 | 33.78 |
| | Gray Hair | 4.75 | 0.28 | 0.73 | 31.50 |
| Cityscapes | Semantic Segmentation | $0.99 \times 10^{-3}$ | 6.93 | -5.32 | -0.39 |
| | Depth Estimation | -0.05 | 0.43 | -6.10 | 2.73 |
| GLUE | STS-B | -0.01 | 189.82 | -73.51 | -1.54 |
| | MRPC | 0.02 | -0.07 | 8.63 | $2.55 \times 10^{-4}$ |
| | RTE | 0.07 | 0.12 | -0.10 | $1.27 \times 10^{-5}$ |
| | SST-2 | 0.01 | 0.16 | 14.14 | 211.80 |

Table 1: Multi-task transfer ($\Delta\mathcal{E}_k = \mathcal{E}_k(\Theta_{MT}) - \mathcal{E}_k(\Theta_{ST})$) for the tasks in each setting that experience *significant impact to generalization*, along with the change to factors detailed in §4. Shading indicates a canonically  positive / negative / insignificant  delta, where insignificance is determined by overlap of 2 standard deviations. For a factor of optimization to potentially explain multi-task transfer, we must see  positive transfer  connected to a  positive  change, and  negative transfer  connected to a  negative  change for that factor across an MTL setting. However, we instead see that, across many multi-task settings, multi-task learning results in an insignificant, or worse, change to factors of optimization regardless of its effect on generalization. **This shows that the trade-off in generalization between tasks in a given MTL setting is not explained by a corresponding trade-off in any factor of optimization trajectories.**

the trace of the gradient covariance matrix:

$$\text{Coherence}\,(\theta) = \underset{(x,y)\in S^k}{\mathbb{E}}\left[(\nabla_\theta\ell(f_\Theta(x),y) - \nabla_\theta\mathcal{L}_k(\Theta))^2\right] \tag{4}$$

Geiping et al. (2022) and Novack et al. (2023) reformulate this bias as a penalty on the gradient norm of *small batches*, and show that explicitly optimizing small-batch gradient norms during large-batch training can recover the generalization gap between large- and small-batch models.

**Early Phase Fisher Information:** Finally, Jastrzebski et al. (2021) found that (in single-task learning) the trace of the Fisher Information Matrix (FIM)

$$\text{FIM Trace}\,(\theta) = \underset{x\in S_k,\hat{y}\sim f_{\theta,\phi_k}(x)}{\mathbb{E}}\left[||\nabla_\theta\ell(f(x),\hat{y})||_2^2\right] \tag{5}$$

in the early stages of training is correlated with final solution generalization. They show that optimization trajectories which maintain a low FIM trace in the early phase of training yield much better generalization than trajectories whose FIM trace "explodes" at the beginning of training. This finding is corroborated by Novack et al. (2023), who show that directly optimizing a small-batch FIM norm leads to better generalization for large-batch training (similar to gradient coherence).

### 4.1 Factors of the Optimization Trajectory are Not Correlated with Trade-Offs in Generalization

In Figure 3 we plot each factor by the total training loss for FashionMNIST in the FashionMTL setting, where the color of each curve corresponds to the generalization trajectory shown in Figure 1. Because we observe that MTL with additional FashionMNIST data (green curve) leads to positive transfer while MTL with NoisyFashion data (red curve) leads to negative transfer, we expect to see that one of these factors is minimized *worse* by the red curve than the single-task curve (blue), and minimized *better* by the green curve. However, we instead find that both MTL trajectories exhibit either better or comparable optimization of each factor when compared to single-task trajectories, despite significant differences to generalization. In other words, the impact of MTL on these factors does not explain negative *and* positive transfer.

In Table 1, for the remaining MTL settings, we compare change in generalization to change in optimization factor for each task that experiences significant transfer, computing an aggregate value of each attribute.[7] The $\Delta$ of each term is computed using the average of all multi-task trajectories (averaged over random seeds) minus the average of all single-task trajectories and the significance of a $\Delta$ is determined by whether the confidence intervals (using 2 standard deviations) of the multi-task and single-task values overlap. Our hope is that one of the factors we study will explain the trade-offs in generalization between tasks, i.e. that positive transfer will correspond to a decrease in one or more factor while negative transfer corresponds to an increase. Such a result would suggest *how* multi-task learning is impacting generalization, and would potentially provide a path towards developing optimization methods that are "right for the right reasons".

However, we find that no factor is capable of explaining negative and positive transfer in any MTL setting we study. For instance, in CIFAR-100, MTL consistently leads to sharper solutions for all tasks; however, sharpness cannot explain MTL transfer because even tasks which experience *positive transfer* find sharper solutions. Other factors not only experience deltas which are not significant, but which are also inconsistent. More generally, we see that multi-task learning tends to result in *worse* values for the factors we study, regardless of whether task generalization is improved or harmed by multi-task training (with the exception of the FashionMTL setting, where MTL seems to improve some factors). This negative result indicates that our current understanding of generalization gaps in *single-task models* is not capable of explaining the generalization gaps between multi-task and single-task models. More importantly, while there is clearly a trade-off in generalization between tasks within some multi-task settings, we cannot explain *why* that trade-off occurs from the perspective of optimization. As a result, it is not clear how we should alter optimization to improve or balance transfer across tasks or, more specifically, what optimization challenges SMTOs should be overcoming to improve multi-task transfer.

> We find that factors of the optimization trajectory previously shown to explain generalization gaps between single-task training runs are not capable of explaining the generalization gaps between single-task and multi-task models. In other words, our current understanding of how properties of optimization dictate generalization in deep learning cannot explain how multi-task learning improves generalization for some tasks while harming others.

### 4.2 Can Factors of the Optimization Trajectory Explain the Impact of SMTOs?

Although we find, in §4.1, that differences between single-task and multi-task generalization are not explained through the lens of optimization, this does not imply that SMTOs have no role to play in deep multi-task learning. Namely, SMTOs do not necessarily seek to align multi-task trajectories with single-task trajectories, but rather aim to address optimization challenges *within a fixed multi-task problem*; in other words, SMTOs aim to improve optimization relative to the uniform multi-task gradient (UMTG, Equation 2), rather than single-task learning. In this section, we compare the training trajectories of SMTOs to those of the UMTG

---

[7]To compute a single-value of each term for a single training trajectory we average values across the training trajectory, using the following heuristics: generalization is computed as the average test-performance of the top-10 validation checkpoints; sharpness is computed as the average sharpness of the last 20 checkpoints at the end of training; gradient covariance is computed as the average covariance of the last 20 checkpoints at the end of training; finally, the FIM Trace is computed as the average of the max 20 values of FIM Trace value (capturing the "explosion"). Full trajectories are shown in Appendix E.

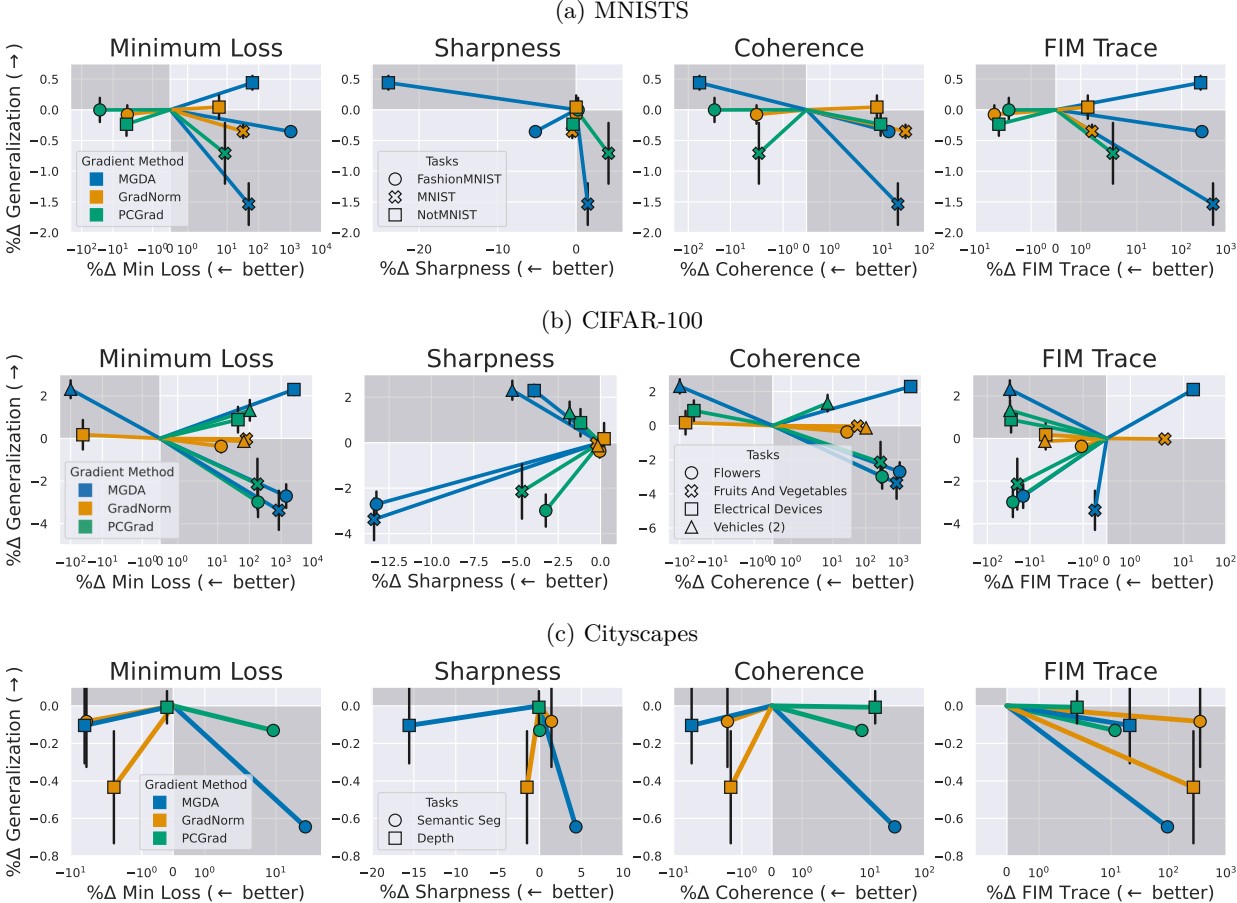

Figure 4: The impact of SMTOs on generalization vs. their impact on optimization trajectories, as their %Δ over the UMTG trajectory. SMTOs aim to impact task generalization by affecting optimization, so we expect to see positive (negative) changes to task generalization are corroborated by positive (negative) changes to at least one factor of optimization. In other words, for a factor to explain how an SMTO impacts generalization, all of an SMTOs points should exist within the shaded quadrants of a plot. However, there is no SMTO whose impacted tasks exist solely in the shaded regions, suggesting that the mechanisms by which SMTOs improve or harm task performance are not explained by task optimization trajectories.

and ask whether, when SMTOs improve (or harm) task performance compared to the UMTG, that their effect is corroborated by an improvement (or harm) to aspects of task *optimization*.

We select 3 SMTOs and compare their training trajectories to the trajectories of the UMTG: MGDA (Sener & Koltun, 2018), PCGrad (Yu et al., 2020), and GradNorm (Chen et al., 2018).[8] We focus on 4 factors of optimization trajectories: in addition to sharpness, early-stage FIM, and gradient coherence (as in §4.1), we also compare the minimum training loss achieved by each trajectory, a factor that is classically used to motivate many SMTOs. For each SMTO and factor, we measure the *percentage change* (%Δ) over the UMTG for each task that experiences a significant impact to generalization; we calculate the percentage change to keep all factors on the same scale across tasks, which may otherwise exist on different orders of magnitude.

---

[8]Each of these methods is representative of a class of SMTOs: MGDA, similar to CAGrad (Liu et al., 2021a) and NashMTL (Navon et al., 2022), is motivated by Pareto-Optimality, and alters the gradient of each step such that convergence to a Pareto-Stationary point is guaranteed; PCGrad, similar to GradientVaccine (Wang et al., 2021) and IMTL-G(Liu et al., 2021b) directly alters the gradient direction of each step such that the resulting step is sufficiently aligned with all task gradients; finally, GradNorm, similar to IMTL-L (Liu et al., 2021b) and RotoGrad-Scale (Javaloy & Valera, 2022) attempts to scale the rate at which tasks are learned, such that all tasks are minimized at an equal rate.

In each plot, we  shade  the two quadrants which correspond to either a simultaneous improvement to optimization and generalization $(-x, +y)$ or a simultaneous degradation to both optimization and generalization $(+x, -y)$. SMTOs aim to influence generalization by improving or balancing properties of optimization, so we expect that, for each SMTO, there is some factor of optimization for which all points (all tasks that the SMTO positively or negatively impacts) exists within the shaded regions, implying a connection between that factor and the mechanisms by which the SMTO improves or trades-off task generalization.

In Figure 4 we plot these comparisons for the FashionMTL, CIFAR-100, and Cityscapes settings.[9] We find that no factor has points which exist solely within the shaded quadrants, i.e. it is not clear what aspects of optimization are actually effected by SMTOs to impact multi-task performance. Of particular note is the minimum training loss, which is of primary concern to many multi-task optimizers; not only do SMTOs often lead to *higher* minimum training loss than the UMTG, but tasks whose minimum training loss is improved by SMTOs do not necessarily generalize better. For example, PCGrad in CIFAR-100 results in worse minimum training loss, for the 4 tasks it impacts, and *better* sharpness and FIM Trace than the UMTG *regardless of whether task performance is improved or harmed by PCGrad optimization.* Kurin et al. (2022) empirically and theoretically demonstrate that many SMTOs induce an early-stopping behavior and suggest that this may explain why SMTOs sometimes outperform the UMTG; here, we do corroborate the notion that SMTOs can result in an early-stopping for several tasks. However, we also show that the early-stopping behavior of SMTOs has little correlation to task improvement, i.e. it does not explain the benefits of SMTOs.

> When SMTOs *do* have an impact on generalization (either positive or negative), that impact is not explained by a corresponding improvement (or harm) to a task's optimization trajectory. Thus, it is not clear what aspects of optimization SMTOs are impacting to have an effect on multi-task transfer.

## 5 Does Gradient Conflict Explain Impact to Optimization or Generalization?

So far, we have focused our analysis on the *trade-offs* between different tasks within a fixed multi-task problem. While we often observe a trade-off in generalization between tasks in an MTL problem, we show that it is not clear from an optimization perspective *why* that trade-off occurs. However, despite being unable to explain the impacts to generalization, multi-task learning—and consequently, gradient conflict— has a significant impact on the optimization trajectories of all tasks. One potential explanation of our result is that multi-task learning pulls optimization into regions of the loss surface that single-task trajectories do not explore, such that comparisons between single-task and multi-task trajectories uninformative.

In this section, we ask whether the *amount* of gradient conflict between a given task and the multi-task gradient drives the impact of MTL on the task's optimization trajectory, and whether this impact is correlated with task generalization. Intuitively, the higher the gradient conflict—i.e., the lower the similarity between the target task gradient and the multi-task gradient—the higher the impact of MTL on optimization may be, as the optimization path diverges more severely from the single-task gradient. We focus our analysis on a few target-tasks from CIFAR-100, which has a large number of auxiliary tasks to choose from; we randomly select sets of auxiliary tasks for 2, 5, 10, and 20-task settings (covering a range of multi-task sizes) and train models with each target-task and auxiliary task set. Changing the set of auxiliary tasks impacts both the amount of gradient conflict that our target-task experiences during training, as well as the ultimate generalization of the model on our target-task. To measure gradient conflict at each epoch of training, we measure the cosine similarity between the target-task gradient and the UMTG gradient:

$$\text{Gradient Similarity}(k, K, \Theta) = \texttt{cosine-sim}(\nabla_\theta \mathcal{L}_k^B, \nabla_\theta^{MT}(\Theta, B)) \tag{6}$$

where $B$ is the batch-size used during training, $\nabla_\theta^{MT}$ is the multi-task gradient (Equation 2), and $\theta$ is all parameters of the shared encoder concatenated into a single vector. To smooth out the impact of noise due to small batch-sizes, we compute the mean gradient similarity over 200 randomly sampled batches. If an

---

[9]As in Xin et al. (2022) and Kurin et al. (2022), we find that most tasks do not experience a significant shift to generalization from SMTOs over the UMTG. In CIFAR-100, we show only tasks that experience a significant change. For Cityscapes, we show both tasks. Finally, for FashionMNIST, we show the 3 tasks that have the largest impact.

auxiliary task gradient has a high amount of conflict with the target-task, the UMTG will be pulled away from the target-task gradient. Thus, the similarity between the target-task gradient and the UMTG captures the average amount of conflict between the target-task and all auxiliary tasks.

## 5.1 Conflict Has a Predictable (Negative) Impact to Optimization Trajectories

We begin by asking whether the similarity of the target-task gradient to the MTL gradient (Equation 6) is indicative of how multi-task training will impact the optimization trajectory. While it is clear that high gradient conflict will have a negative impact on the minimization of the training loss (Nocedal & Wright, 2006; Yu et al., 2020), we have seen in §3 that the multi-task transfer cannot be solely explained by the impact of gradient conflict on the training loss. Thus, we ask here whether a high amount of conflict between the target-task and the multi-task gradients has a negative impact on the factors of optimization trajectories that we study in §4. For each target-task, we compare these factors (early-stage FIM, sharpness, and gradient coherence) with the average gradient similarity between the target-task gradient and the UMTG throughout training, and plot the results in Figure 5a.

Overall we find that there is a *positive correlation* between the amount of gradient conflict a target-task experiences, and the value of the factors we study; as the similarity between the target-task and multi-task gradient decreases, sharpness, early-stage Fisher information, and gradient covariance *all* increase correspondingly during training. While prior work has shown that high conflict can create problems for zeroth order training loss minimization, e.g. poor local minima (Yu et al., 2020) or failure to optimize all tasks simultaneously (Chen et al., 2018), here we show that high gradient conflict negatively impacts more than just the training loss trajectory, including factors which have been previously tied to generalization. However, while it is clear that gradient conflict has a predictable impact on optimization, it is not clear whether this is a *problem* from the perspective of transfer; in other words, it is not clear whether the impact to target-task optimization is related to the target-task's generalization.

> High gradient conflict *negatively impacts* task optimization in more ways than just training loss minimization. We find that the impact of MTL on task optimization—including factors such as the sharpness, Fisher information, and gradient coherence—is correlated with the amount of gradient conflict that the task experiences during the early stages of training.

## 5.2 Conflict Does Not Have a Predictable Effect on Generalization

While some amount of conflict between tasks is intuitively necessary for MTL to improve generalization over single-task learning (Du et al., 2020), the intuition behind many SMTOs is that a *high* amount of gradient conflict can negatively impact task optimization and generalization (Yu et al., 2020; Chen et al., 2020; Liu et al., 2021a, *inter alia)*. Indeed, in §5.1 we see that higher amounts of gradient conflict can have a negative impact on training trajectories, worsening several factors associated with generalization gaps. However, as we have shown in §3, MTL having a negative impact on training loss optimization does not necessarily imply a negative impact on task generalization. To that end, we ask whether the amount of conflict that a target-task experiences during training is predictive of its generalization after training.

We plot target-task test accuracy by gradient similarity for our 3 CIFAR-100 target-tasks in Figure 5b. Unlike factors of the optimization trajectory, we see that the gradient similarity between the target-task and the multi-task gradient has very little correlation with target-task generalization. In fact, for the target-task Fish, the auxiliary setting with the lowest gradient similarity to the target-task (i.e. for which optimization consistently moves further away from the steepest single-task direction of descent) yields the *strongest generalization* of all auxiliary settings, despite leading to higher FIM explosion, higher gradient covariance, and higher sharpness than the other settings. Based on the results of §5.1, intuition would suggest that we should mitigate gradient conflict (i.e. maximize the similarity of our gradient step with the gradient of our target-task) to improve training loss minimization, sharpness, gradient coherence, and minimize FIM explosion. However, our results in Figure 5b demonstrate that while mitigating conflict can improve many aspects of task *optimization*, it will not necessarily improve task *generalization*.

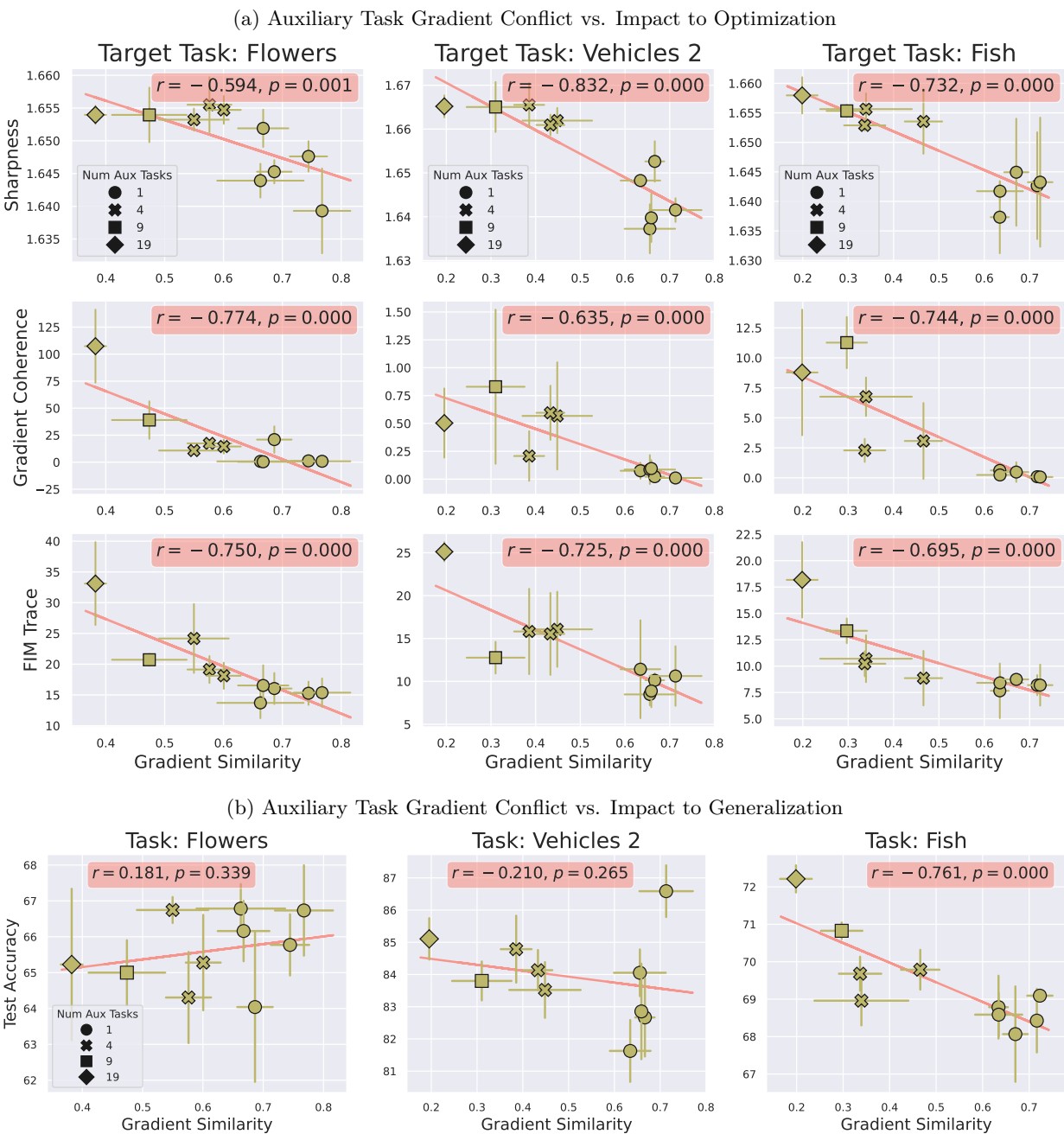

Figure 5: The impact of gradient conflict on factors of the target-task optimization (a) and generalization (b) across auxiliary task settings. Pearson-r correlation coefficient and p-value are shown at the top. Gradient similarity is negatively correlated with each of the optimization factors that we study, implying that *high gradient conflict negatively impacts many factors of task optimization*; however, gradient conflict is *not* negatively correlated with target-task generalization. **In other words, while gradient conflict has a consistent, negative impact to optimization, this effect does not predict or explain transfer.**

> Although early-stage gradient conflict has a predictable impact on target-task optimization (§5.1), this correlation does not hold for *generalization*. It is therefore not clear why mitigating gradient conflict—for instance, via specialized optimizers that manipulate the task gradients—should improve multi-task transfer.

# 6 Conclusion

We empirically show, across a number of multi-task settings, that both positive and negative transfer are determined *early into training*, implying that negative transfer is a result of gradient conflict impacting factors of optimization that go beyond the zeroth order training loss. To understand how gradient conflict causes negative transfer for some tasks while benefiting others, we study factors of optimization that have previously been shown to explain generalization gaps in single-task learning. Our goal is to show how trade-offs in transfer can be explained as trade-offs in task optimization. However, we find that *no* factors we study can adequately explain the impact of multi-task learning on task generalization, i.e. it is not clear how the impact of task conflict on optimization is related to negative (and positive) transfer. Moreover, we find that when SMTOs impact which tasks benefit and which are harmed within a multi-task setting, this impact is not explained by a corresponding improvement or degradation in the tasks' optimization, *including how well the task losses are minimized.* Finally, we study how the *amount* of gradient conflict impacts transfer and we find that high amounts of gradient conflict are highly correlated with h, while having almost no correlation with task generalization.

Overall, we show that predicting which tasks will experience positive transfer and which will experience negative transfer, through the lens of optimization alone, is currently not feasible. This result not only explains why current SMTOs—motivated by gradient conflict and training loss minimization—fail to generally help in practice, but also rules out several additional factors of training trajectories that might, in theory, be used to predict transfer. Our results also have implications for the development of future SMTOs, such as narrowing the focus of future SMTOs to the early phases of training where the generalization gap occurs.

**Future Directions** Our work raises several open questions around transfer in deep learning: Can the benefits and failures of MTL be explained through the lens of optimization? Can multi-task transfer be understood through a single mechanism, as many SMTOs claim, or does it vary across model and setting? And, can multi-task transfer within a given model be improved, over the UMTG, by addressing specific aspects of optimization? Answering these questions is crucial towards not only improving multi-task models in practice, but also towards understanding how models generalize in deep learning in general. Our work also suggests that, without answers to the above questions, the most principled approach to addressing failures in multi-task learning may not be to focus on optimization in deep learning, but to instead focus on properties of the tasks themselves, e.g. meta-learning transfer from task distributions as in Song et al. (2022).

**Limitations** As in any empirical analysis such as ours, our takeaways are limited by the number of settings and the breadth of models that we consider. While we mitigate this by considering 5 unique MTL settings and performing extensive hyperparameter searches, there remains the possibility that our results do not generalize to other tasks or domains. Additionally, our takeaways are naturally limited by the factors of optimization that we study, and their capacity to explain generalization in deep learning. While we can rule out the potential for certain aspects of optimization to explain transfer, it remains an open question as to whether or not *some* aspect of optimization can predict the benefits and failures of MTL in deep learning.

# Acknowledgements

We would like to sincerely thank Steven Reich for his contributions to, and helpful discussions of, an earlier version of this work. We would also like to thank Michael Crawshaw, Neha Verma, Desh Raj, and Suzanna Sia for their feedback on recent drafts of this paper, as well as Sophia Hager, Rachel Wicks, Aleem Khan, Rafael Rivera-Soto, Andrew Wang, Ashi Garg, and Cristina Aggazzotti. Finally, we would like to thank the anonymous reviewers who's helpful feedback improved our work. This work was supported, in part, by the Human Language Technology Center of Excellence at Johns Hopkins University.

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

# A    Discussion of Limitations, Broader Impact, and Future Work

**Future Directions**   Our work raises several open questions around transfer in deep learning: Can the benefits and failures of MTL be explained through the lens of optimization? Can multi-task transfer be understood through a single mechanism, as many SMTOs claim, or does it vary across model and setting? And, can multi-task transfer within a given model be improved, over the UMTG, by addressing specific aspects of optimization? Answering these questions is crucial towards not only improving multi-task models in practice, but also towards understanding how models generalize in deep learning in general. Our work also suggests that, without answers to the above questions, the most principled approach to addressing failures in multi-task learning may not be to focus on optimization in deep learning, but to instead focus on properties of the tasks themselves, e.g. meta-learning transfer from task distributions as in Song et al. (2022).

**Broader Impact**   The goal of this paper is to deepen our understanding of how neural networks learn to generalize when learning jointly from diverse signals. Understanding the mechanisms by which multi-task learning impacts generalization has both practical implications (for a broad array of real-word settings that use multi-task training) as well as theoretical implications (by uncovering, corroborating, or contradicting explanations of how generalization is connected to optimization in deep learning). Furthering this understanding has important societal implications, such as understanding what types of distributions are harmful for models or lead to certain behaviors, or allowing the construction of more interpretable ML systems.

**Limitations**   As in any empirical analysis such as ours, our takeaways are limited by the number of settings and the breadth of models that we consider. While we mitigate this by considering 5 unique MTL settings and performing extensive hyperparameter searches, there remains the possibility that our results do not generalize to other tasks or domains. Additionally, our takeaways are naturally limited by the factors of optimization that we study, and their capacity to explain generalization in deep learning. While we can rule out the potential for certain aspects of optimization to explain transfer, it remains an open question as to whether or not *some* aspect of optimization can predict the benefits and failures of MTL in deep learning.

## B   Dataset & Model Details

- **FashionMTL:** FashionMTL is a synthetic setting where we consider the impact of training a task jointly with (a) an ideal task where we see positive transfer versus (b) an uninformative task which yields negative transfer. Our target-task in this setting is the FashionMNIST task (Xiao et al., 2017), which is a 10-class image classification task where the goal is to classify images of clothing articles. We randomly split the FashionMNIST dataset into two splits of $25,000$ samples each, and consider the first split to be our target-task training dataset. The second half of the dataset is used for the auxiliary task training: for setting (a)—the ideal auxiliary task—we use the remaining $25,000$ FashionMNIST samples as the auxiliary task with no perturbations. In this setting, our auxiliary task is additional data from the exact same distribution of our target-task training data. For setting (b), we use the same remaining $25,000$ FashionMNIST samples, but we randomize the labels. In setting (b) our auxiliary task is now uninformative, as it amounts to fitting noise, and conflicts with the objective of our target-task. For each task we have $5,000$ validation and test samples. Our architecture is a simple LeNet CNN architecture whose penultimate representation is passed to task-specific linear classification layers.

- **MNISTS:** MNISTS (Hsieh & Chen, 2018) is a 3-task multi-task setup involving 3 MNIST-like datasets (all $28 \times 28$ greyscale images). The 3 datasets are MNIST (LeCun et al., 1998), a 10-class handwritten digit recognition task; FashionMNIST (Xiao et al., 2017), a 10-class clothing article classification task; and NotMNINST, a 26-class English character classification task. Each task has exactly $50,000$ training samples and $5,000$ validation and test samples. Our architecture for this setting is a simple LeNet CNN architecture whose penultimate representation is passed to task-specific linear classification layers.

- **CIFAR-100:** CIFAR-100 (Krizhevsky, 2012) is a hierarchical 100-class classification dataset; these class hierarchies can be separated into 20 individual 5-class classification tasks, e.g. Household Electronics classification or Aquatic Mammals classification. Each task consists of $5,000$ training samples (roughly $1,000$ samples per-class), and $500$ validation and test samples. We use a ResNet18 architecture whose penultimate representation is passed to task-specific linear classification layers.

- **CelebA:** CelebA (Liu et al., 2015) is a 40-way binary attribute prediction task. It consists of images of celebrity faces, each of which is labeled with 40 binary attributes. We treat these attributes as separate, binary classificaiion tasks, yeilding a 40-task problem. The dataset is large, Each task consists of $162,700$ training samples with $19,867$ validation and $19,962$ test samples. We use a ResNet18 architecture whose penultimate representation is passed to task-specific linear classification layers.

- **Cityscapes:** The Cityscapes dataset consists of images of urban streets and is cast as a two-task multi-task setting: 7-class semantic segmentation and depth estimation. The dataset consists of $2,975$ train images, $500$ validation images, and $1,525$ test images; we follow the same pre-processing steps used in Sener & Koltun (2018). To model these tasks we use the DeepLabV3 architecture (Chen et al., 2017), which consists of a ResNet101 backbone, pre-trained on ImageNet, and task-specific Atrous Spatial Pyramid Pooling modules.

- **GLUE:** Finally, we consider the GLUE dataset (Wang et al., 2018), a benchmark of 8 NLP tasks.[10] 7 tasks are classification tasks, ranging from Natural Language Inference to Grammatical Correctness, and one task is a regression task (Semantic Similarity). We use a pre-trained RoBERTa-Base (Liu et al., 2020) backbone, with linear task heads on top of the penultimate representation of the [CLS] token, as is standard for BERT fine-tuning (Devlin et al., 2019).

For all settings, we conduct a hyperparameter sweep over the learning rate: $\{10^{-1}, 50^{-1}, 10^{-2}, 50^{-2}, 10^{-3}, 50^{-3}, 10^{-4}, 50^{-4}, 10^{-5}\}$ and batch size: $\{4, 16, 32, 64, 128, 256\}$. We use the Adam optimizer over all settings, as we found it to yield the best generalization in our settings out of { Adam, SGD, SGD with Momentum}. In all settings we use a constant learning rate (no decay).

## C  Calculating Trajectory Sharpness, Fisher Information, and Gradient Coherence

### C.1  Sharpness

We calculate loss surface sharpness using the worst-case $\ell_\infty$ sharpness measure over a batch-size, $|B|$, of 128, following the details and implementation of Andriushchenko et al. (2023). At any given position, $\theta$, in the loss surface, the equation for worst-case $\ell_\infty$ sharpness is

$$\text{Sharpness}\,(\theta) = \mathbb{E}_{B \sim S_k} \max_{||\,|\theta|^{-1}\epsilon||_\infty < \rho} \mathcal{L}_k^B(\theta + \epsilon) - \mathcal{L}_k^B(\theta) \tag{7}$$

where $|B|$ is the batch-size that the loss is calculated over, $S_k$ is the size of the dataset (which we truncate to be of size 2048, following Andriushchenko et al. (2023)) and $\rho$ is a hyperparameter which dictates the maximum size of the perturbations over which the worst-case loss is calculated. We experiment with several different values of $\rho = \{10^{-3}, 5^{-3}, 10^{-2}, 10^{-1}\}$, but we find that this parameter does not change the relative ordering between our different trajectories. In general, we default to using $\rho = 10^{-3}$.

### C.2  Fisher Information

The trace of the Fisher information matrix of a given neural network is given as the average $\ell_2$ norm of the gradient of $\theta$ w.r.t. the distribution over $y$ given by the model:

$$\text{FIM Trace}\,(\theta) = \mathbb{E}_{x \in S_k, \hat{y} \sim f_{\theta, \phi_k}(x)} \left[ ||\nabla_\theta \ell(f(x), \hat{y})||_2^2 \right] \tag{8}$$

In practice, we calculate the FIM Trace over a mini-batch of size $|B| = 16$, as Novack et al. (2023) demonstrate empirically that mini-batch FIM explosion is correlated with the large-batch generalization gap. Additionally, we truncate the size of the datasets $S_k$ to be of size 2048. In most cases, we calculate the expectation by sampling $\hat{y} \sim f_{\theta, \phi_k}(x)$ many times from $f_{\theta, \phi_k}$ many times for each $x$. In cases where the model does not give a natural distribution to exactly sample $\hat{y}$, we approximate instead with the empirical fisher information matrix, using the true label $y$ (for instance, regression tasks such as STS-B or Depth Estimation).

### C.3  Gradient Coherence

Motivated by McCandlish et al. (2018), we approximate the gradient covariance using several small batches and an aggregated large-batch. We compute an *aggregated* large batch gradient of size $|L|$ by first computing $n$ gradients of batch-size $|B| = \frac{|L|}{n}$, and averaging them.

$$\nabla \mathcal{L}^L(\theta) = \sum_{i=1}^{n} \nabla \mathcal{L}^{B_i}(\theta) \tag{9}$$

During the computation of each gradient, we compute and store the gradient norm, giving us $n$ samples of a batch-size $|B|$ gradient norm; we denote the average of these gradient norms as $G_B$. Additionally, after the computation of the aggregated gradient of batch-size $L$, we compute its norm which we denote by $G_L$.

---

[10]We exclude the WNLI task, as it is well-documented to be extremely prone to overfitting (Devlin et al., 2019).

Conveniently, as noted by McCandlish et al. (2018), access to $G_B$ and $G_L$ provides a method to approximate the gradient covariance. Namely, we can write the expected gradient norm of a batch-size $|Z|$ as

$$\mathbb{E}[||\nabla\mathcal{L}^Z(\theta)||^2] = ||\nabla\mathcal{L}(\theta)||^2 + \frac{A_Z}{|Z|}tr(\Sigma(\theta)) \tag{10}$$

where $A_Z$ is the FPC for a batch-size of $|Z|$, $\frac{|S_k|-|Z|}{|S_k|-1}$, because examples are sampled without replacement. We can therefore write

$$\mathbb{E}[||\nabla\mathcal{L}^B(\theta)||^2] - \frac{A_B}{|B|}tr(\Sigma(\theta)) = \mathbb{E}[||\nabla\mathcal{L}^L(\theta)||^2] - \frac{A_L}{|L|}tr(\Sigma(\theta)) \tag{11}$$

and from this we see that

$$tr(\Sigma(\theta)) = (\mathbb{E}[||\nabla\mathcal{L}^B(\theta)||^2] - \mathbb{E}[||\nabla\mathcal{L}^L(\theta)||^2])\frac{1}{\frac{A_B}{|B|} - \frac{A_L}{|L|}} \tag{12}$$

$$\approx (G_B - G_L)\frac{1}{\frac{A_B}{|B|} - \frac{A_L}{|L|}} \tag{13}$$

which gives us an approximation to $\mathcal{B}_{SGD}$. In many cases, the FPC factors $A_Z$ are ignored because $|Z| \ll |S_k|$ and thus $A_Z \approx 1$. However, in some settings, $|L|$ approaches $|S_k|$, and there we do not ignore the FPC.

### C.4 Aggregating Values Over a Training Trajectory

In general, we are interested in the value of these factors over specific points of the training trajectory. For instance, we are interested in how sharpness and gradient covariance are minimized towards the end of training, near convergence; alternatively, we are interested in how the FIM trace explosion is minimized at the initial phase of training. However, these factors are also connected to the magnitude of the training loss (e.g. in practice we see that a low training loss typically has smaller gradients and therefore lower gradient covariance). Therefore, when comparing these factors between different training trajectories, we compare values over *similar loss* by comparing points that fall within a loss bin. For instance, for early-stage FIM Trace explosion, we first compute the values of training loss that constitute the "early phase" of training by creating 20 geometrically spaced bins for a training trajectory, and selecting bins $2-5$. This gives us a loss bin which captures the early phase of training, but ignore the very early points that typically have abnormally high loss; on tasks like MNISTS or CIFAR-100, this often constitutes a bin with a max value of $10^0$ and a minimum value of $10^{-1}$, which is where we typically see the FIM explode.

For sharpness and gradient coherence, which we measure towards the *end* of training, we allow the bin to be set by the training trajectory that has the highest minimum loss. For example, when comparing the sharpness of a multi-task and single-task trajectory, the multi-task trajectory will typically converge to a training loss that is an order or magnitude (or more) higher than the single-task trajectory. In this case, we select a loss bin that is computed to capture the end of the multi-task trajectory, e.g. bins $18-20$ out of 20 geometrically spaced bins over the multi-task trajectory. Then, we compare the values of the single-task and multi-task sharpness that fall within those bins, so that we are always comparing values at similar loss values. We emphasize this because we know, from §3, that multi-task transfer can be seen as a gap in generalization for comparable training loss (often from the early phase of training all the way to convergence). Therefore, it is important to focus our analysis on comparing and contrasting aspects of training trajectories at comparable training loss.

## D Discussion of Related Work

### D.1 Multi-Task Transfer and Optimization

The focus of our work is on understanding how, in deep multi-task learning (specifically, in the shared encoder setting; Ruder, 2017; Crawshaw, 2020), generalization is impacted by the joint optimization of many tasks. In the recent past, many multi-task optimization methods have been proposed to mitigate negative

transfer, balance generalization across tasks, or even improve positive transfer in multi-task training; these methods all operate by directly attempting to tackle a variety of optimization problems caused by gradient conflict (see §2.1). However, while the consequences of gradient conflict on optimization are well established in theory, the impact of gradient conflict on *generalization* in deep learning is much murkier. For instance, in the related area of auxiliary task selection, methods often rely on signals from validation data to select related tasks (Standley et al., 2020; Wang et al., 2020; Jiang et al., 2023), as it has been shown that gradient conflict is not necessarily indicative of auxiliary task benefits (Du et al., 2020; Jiang et al., 2023). Moreover, recent work has also found, in MTL, that the amount of conflict *towards the end of training* has no relevance to generalization, while conflict at the beginning of training is only partially correlated with transfer (Royer et al., 2023). Finally, the efficacy of SMTOs has recently been called into question by Xin et al. (2022) and Kurin et al. (2022), who empirically demonstrate that many SMTOs do not actually improve generalization over the UMTG. These findings all call into question the ability of gradient conflict to explain transfer.

By demonstrating that multi-task transfer occurs as a generalization gap early into training (§3) and studying factors previously used to explain generalization gaps in prior deep learning studies (§4.1), we hope to close the gap in our understanding on the connection between optimization and generalization in deep MTL. However, the negative results of our work instead largely highlight the *inability* of optimization to explain the impact of multi-task learning on generalization. Overall, our results offer a potential explanation to the findings of Xin et al. (2022) and Kurin et al. (2022), e.g. that specialized optimizers do not improve multi-task transfer because it is still unclear how multi-task transfer is impacted by optimization.

An alternative approach to predicting task relationships or transfer is to instead model task relationships in a model-agnostic, or at least optimization-agnostic, way. For instance, Vu et al. (2020) look to generate task-embeddings from a pre-trained model and use those embeddings to compute task similarity. Li et al. (2023) propose to train a surrogate model to predict the relevance of a set of auxiliary tasks given many examples of model performance across auxiliary samples. Meta-learning has also been proposed as an approach to learn the relationships between a group of tasks, using validation data to meta-learn which auxiliary tasks will be beneficial to the targe-task Song et al. (2022); Liu et al. (2022). These works often focus on predicting target-task transfer given a set of source-tasks rather than improving the performance of a multi-task model given a fixed set of tasks or leverage validation data to predict task relevance.

### D.2 Generalization Gaps in Deep Neural Networks

Generalization gaps between training trajectories of identical architectures are perhaps most popular in the "large-batch generalization gap" literature, i.e. the observation that small-batch models generalize better than large-batch models at similar training loss (first noted by Keskar et al., 2017; Smith et al., 2018). Early attempts to explain the large-batch generalization gap relied on the width, or sharpness, of the solutions found by large versus small-batch training (Hochreiter & Schmidhuber, 1997), and prior work speculated that the level of noise in the gradient estimates dictated the flatness of the final solution (Jastrzębski et al., 2018; Smith et al., 2020; Li et al., 2021). However, the connection between sharpness and generalization has recently been called into question, with several works finding that metrics for sharpness are not necessarily correlated with generalization (Kaur et al., 2023; Andriushchenko et al., 2023; Mueller et al., 2023) and that sharpness is instead correlated with the learning rate used during optimization (Cohen et al., 2021; Kaur et al., 2023). In response, recent work has explored alternative explanations for generalization gaps; notably, Jastrzebski et al. (2021) find that generalization may be dictated by the maximum trace of the Fisher Information Matrix (FIM) in the *early stages* of training, for which small-batch models have lower explosion. Separately, Smith et al. (2021) leverage backwards analysis to show that mini-batch SGD optimizes a modified loss that contains an implicit bias towards small gradient covariance. Both the Fisher Information Trace and Gradient Covariance have been empirically shown to improve generalization when explicitly optimized in large- or full-batch training (Geiping et al., 2022; Novack et al., 2023).

Notably, attempts to explain the large-batch generalization gap have focused on how aspects of *optimization trajectories*—e.g. how the surface sharpness is impacted, or how well gradient coherence is implicitly minimized—may explain why trajectories generalize differently. In this work, we demonstrate that multi-task transfer (both negative and positive) elicits a generalization gap, similar to the large-batch generalization gap in single-task learning (§3). However, we find that the aspects of optimization previously tied to gener-

alization gaps are unable to explain the generalization gaps we observe in multi-task learning, both between single-task and multi-task trajectories (§4.1, §5.2) and between the trajectories of different optimization methods in a fixed MTL problem (§4.2). Our findings not only empirically demonstrate the difficulty of explaining transfer in deep learning, but also emphasize a disconnect between how gradient conflict impacts optimization versus how gradient conflict impacts generalization in deep multi-task learning.

## E   Full Task Results

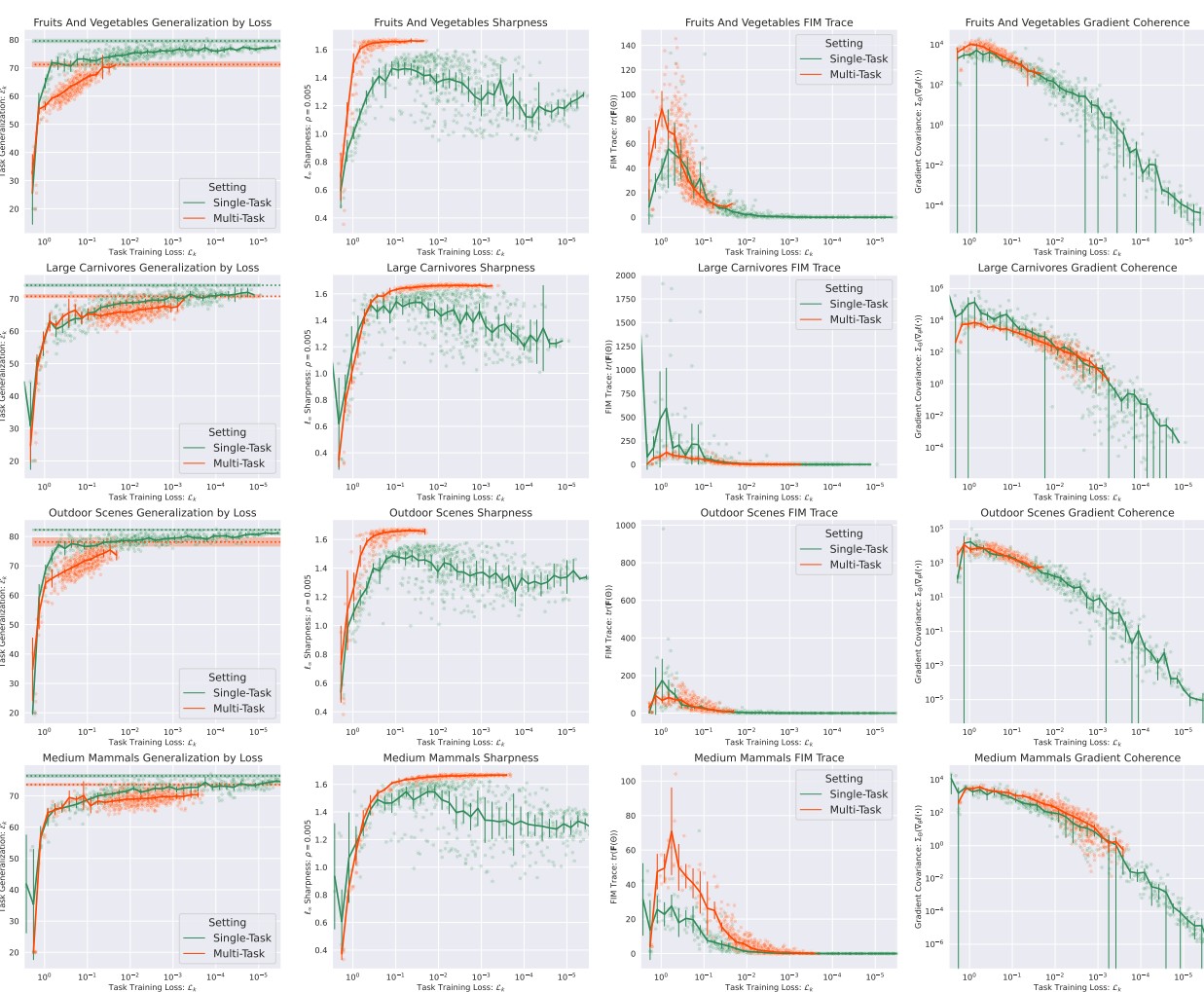

Figure 6: CIFAR Tasks which experience negative transfer.

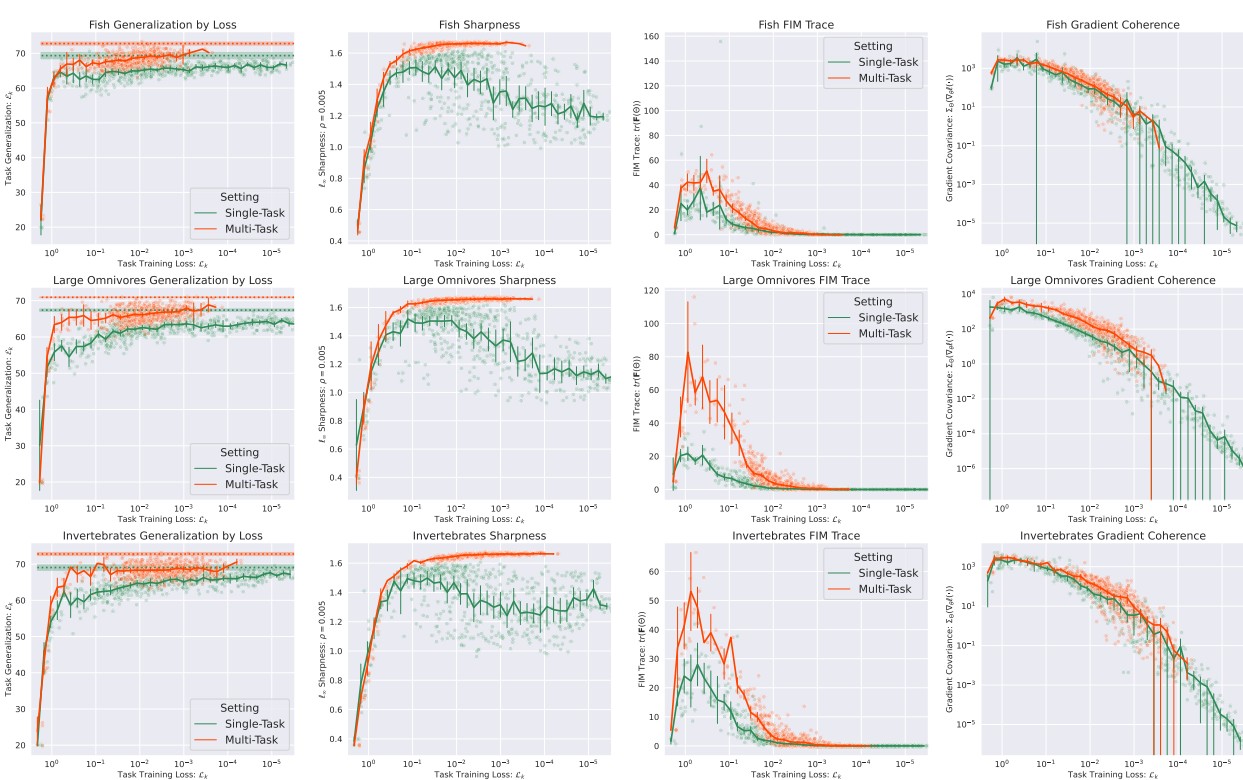

Figure 7: CIFAR Tasks which experience positive transfer.

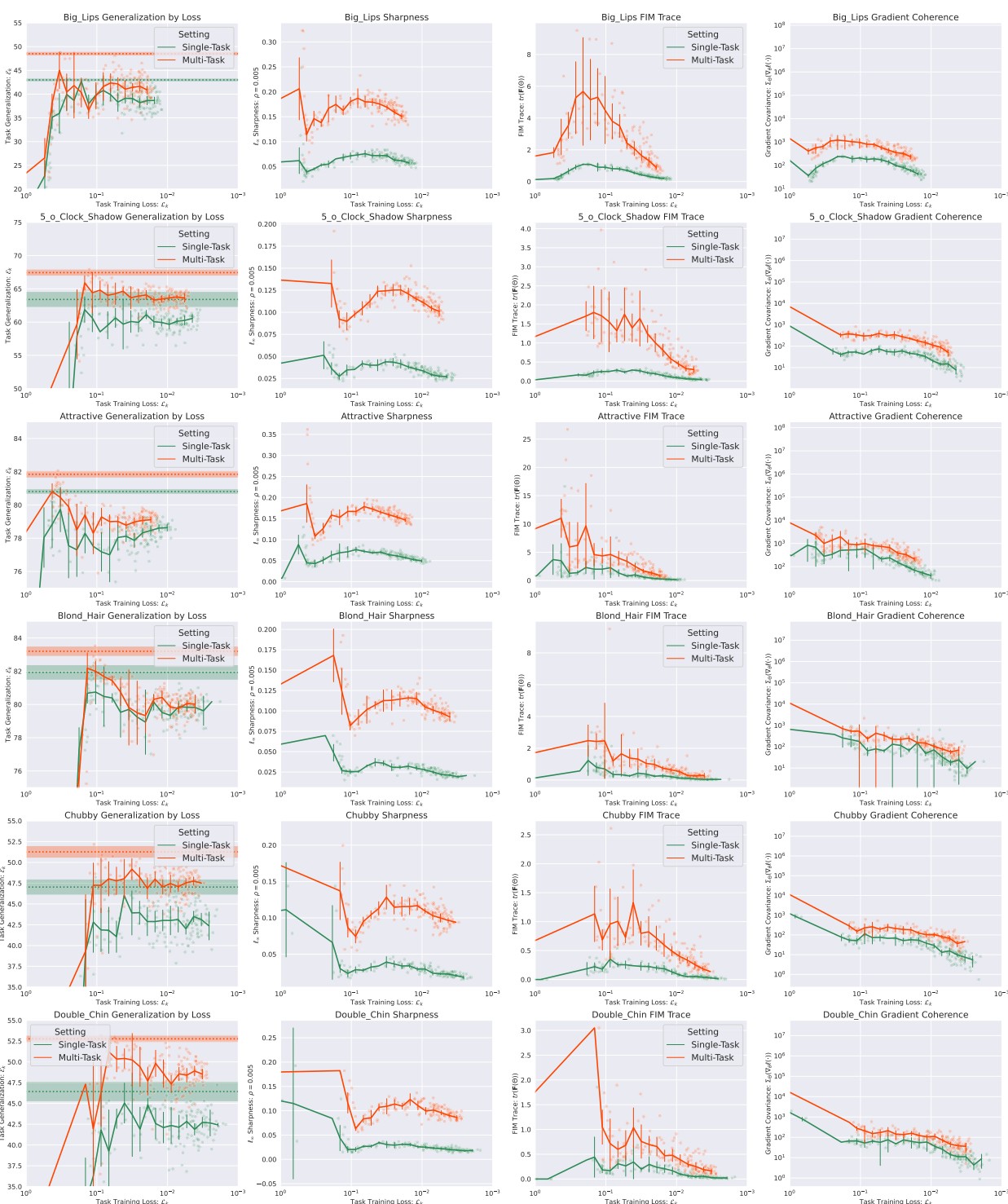

Figure 8: CelebA Tasks which experience significant transfer.

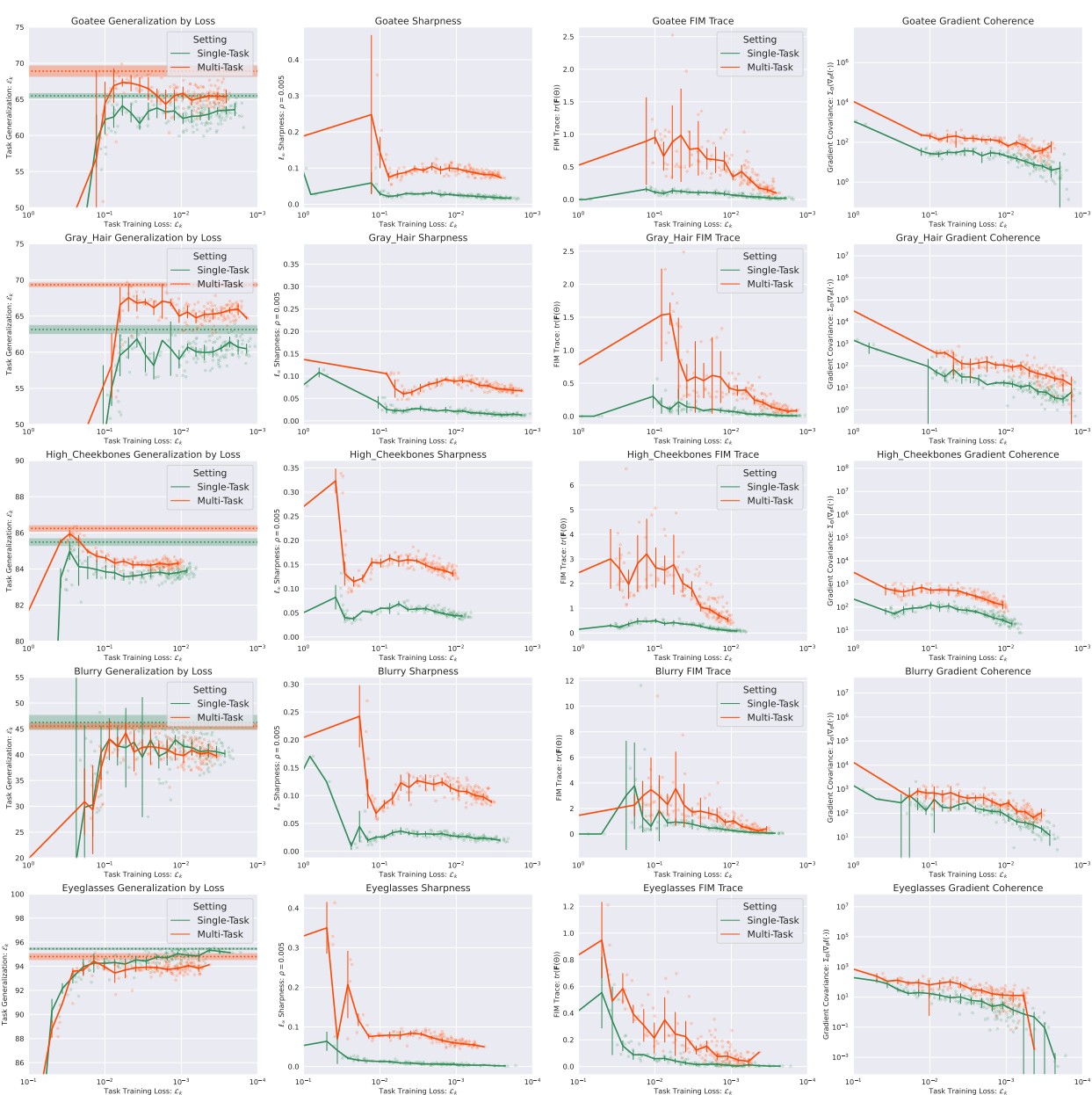

Figure 9: CelebA Tasks which experience significant transfer.

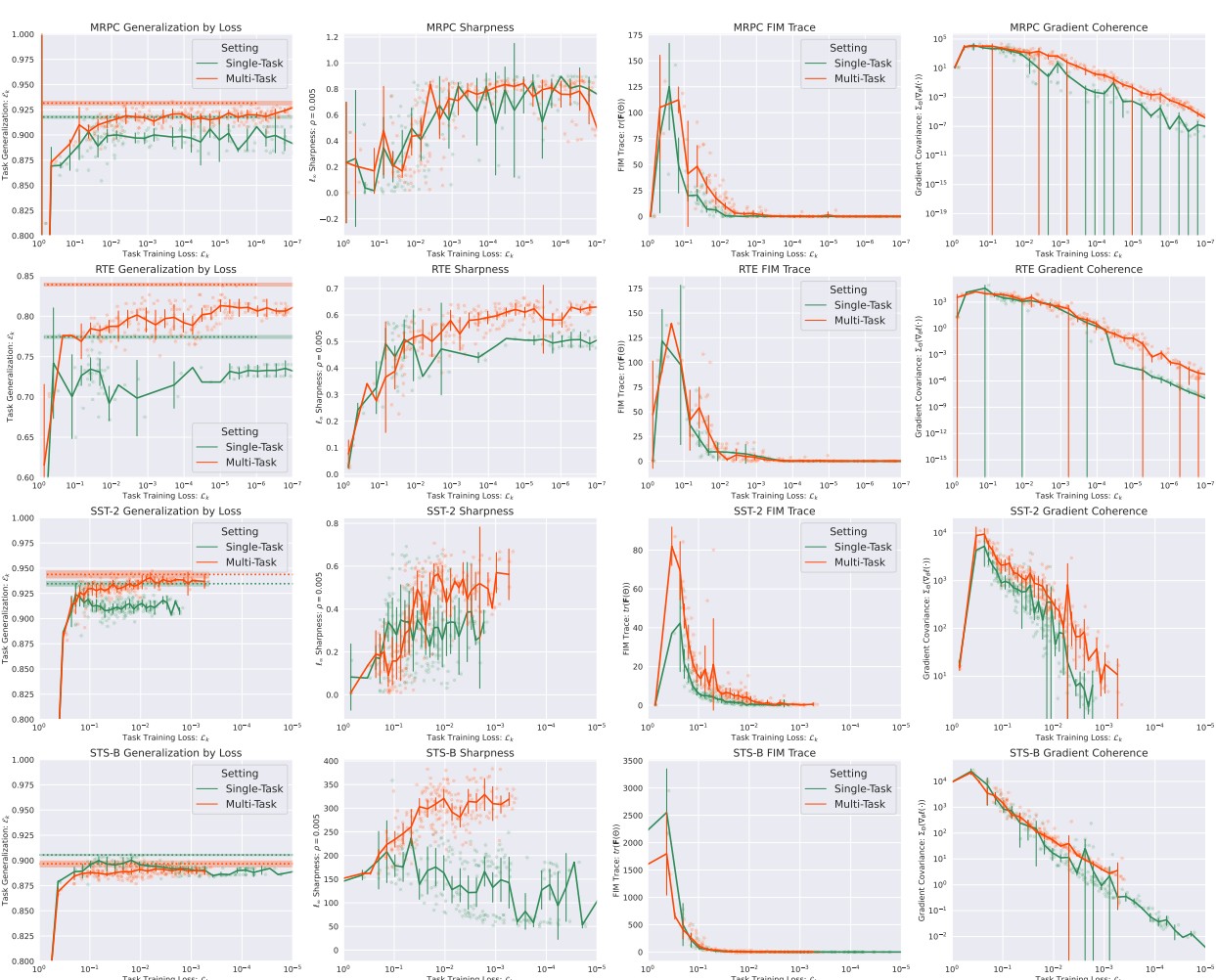

Figure 10: GLUE Tasks which experience significant transfer.

