# OpenReview forum: "Can Optimization Trajectories Explain Multi-Task Transfer?"
_TMLR — Accepted by TMLR_

### Review · Reviewer_yNT4 · 2024-10-12

**Summary Of Contributions:**

- This paper conducts extensive experiments on six multitask datasets (FashinMNIST, MNIST, CIFAR100, CelebA, Cityscapes, GLUE), and provides abundant results on the task transfer results and the evaluations of model optimization trajectories, such as sharpness, Fisher, and gradient covariance.

- This paper shows that the generalization gap between single-task and multitask optimization trajectories emerges in the early phase of training.

- This paper shows that no single factor from the optimization trajectory (sharpness, Fisher, gradient covariance, gradient cosine similarity) can predict the multitask transfer results.

**Audience:**

Yes

**Claims And Evidence:**

Yes

**Requested Changes:**

- For Figure 1, is the x-axis the training loss on a single target task or multiple tasks? If it is the latter, what does it mean when it reaches the same total training loss for STL and MTL? It would be better to explain more in the figure's title.

- In Section 3, the paper claims that negative transfer incurs generalization gaps between STL and MTL trajectories comparatively early into training. How early does the negative transfer start to appear in training? It would be helpful to show figures of training/text losses w.r.t. number of epochs or iterations.

- In Section 4, the paper observes that no factor is capable of explaining negative and positive transfer in any MTL setting we study. What is the correlation between the factors to negative transfers? Can we also measure the factors at the checkpoint at each epoch to study the correlation? How about considering various subset combinations of tasks (such as in GLUE), and how do the factors correlate with the transfer results (it would be better to report correlation scores)?

- I also have similar questions about the results in Figure 4. It seems that there is some correlation between the factors and the transfer results. It remains unclear **how strong/weak the correlation is**. For example, can one fit a linear or polynomial model using the values of factors to predict the transfer results? How significant would that prediction be? It would be better to show similar figures like Figure 5 or 6.

- It would be better to provide a section to describe the practical implications of the observations: such as how can we choose MTL optimization algorithms given the evaluated factors in the early phase of training (sharpness, Fisher, gradients)?

- There are prior works on predicting negative transfers in multitask learning, which should be discussed in the paper:

[1] Vu, T., Wang, T., Munkhdalai, T., Sordoni, A., Trischler, A., Mattarella-Micke, A., ... & Iyyer, M. (2020). Exploring and Predicting Transferability across NLP Tasks. In Proceedings of the 2020 Conference on Empirical Methods in Natural Language Processing (EMNLP)

[2] Li, D., Nguyen, H., & Zhang, H. R. (2022) Identification of Negative Transfers in Multitask Learning Using Surrogate Models. Transactions on Machine Learning Research.

**Strengths And Weaknesses:**

### Strengths

- This paper conducts extensive experiments on six multitask datasets (FashinMNIST, MNIST, CIFAR100, CelebA, Cityscapes, GLUE), and provides abundant results on the task transfer results and the evaluations of model optimization trajectories, such as sharpness, Fisher, and gradient covariance.
- This paper shows that no single factor from the optimization trajectory (sharpness, Fisher, gradient covariance, gradient cosine similarity) can predict the multitask transfer results.

### Weaknesses

- This paper does not have a principled explanation for the empirical results. There are many empirical results of the evaluations in optimization trajectories. Yet, the conclusion from the results is that no single factor can predict the transfer results. The practical implication is limited.

---

> ### Author Response · Authors · 2024-11-21
> **Response to Reviewer yNT4**
>
> Thank you for your review! We would like to begin by addressing the weakness listed:
>
> > This paper does not have a principled explanation for the empirical results. There are many empirical results of the evaluations in optimization trajectories. Yet, the conclusion from the results is that no single factor can predict the transfer results. The practical implication is limited.
>
> While it is true that our take-away is that no single factor can predict multi-task transfer, we strongly disagree that the practical implication is limited. Nearly every SMTO is grounded in a proof or demonstration in a toy setting showing that the SMTO achieves stronger convergence or lower training loss than its competitors and the UMTG. Our results therefore have major implications for this field, as we empirically demonstrate that these toy settings and theoretical assumptions should not be used as evidence that a method will improve transfer. Moreover, while many of these SMTOs have been called into question by [1] and [2] who empirically demonstrate that these results do not significantly improve transfer, our results explain why they fail to improve transfer and moreover show that many assumptions still held by the multi-task community are not accurate in multi-task deep learning.
>
> Finally, our observation that transfer arises early into training has implications for where we should look to improve multi-task transfer in deep learning when developing future methods, and we further rule out several hypotheses for what factors may explain how transfer is impacted early into training. Therefore, we argue that we have laid the groundwork for future methods based on our new understanding of multi-task optimization dynamics.
>
> [1] Kurin et al., 2022; In Defense of the Unitary Scalarization for Deep Multi-Task Learning
>
> [2] Xin et al., 2022; Do Current Multi-Task Optimization Methods in Deep Learning Even Help?
>
> Regarding the requested changes:
>
> > For Figure 1, is the x-axis the training loss on a single target task or multiple tasks? If it is the latter, what does it mean when it reaches the same total training loss for STL and MTL? It would be better to explain more in the figure's title.
>
> In figure 1 the x-axis is the training loss for a single task (Fashion1). Each line and hue plots the training loss by test accuracy of Fashion1 when trained in distinct MTL or STL settings. We have updated the caption of Figure 1 to make this more clear.
>
> > In Section 3, the paper claims that negative transfer incurs generalization gaps between STL and MTL trajectories comparatively early into training. How early does the negative transfer start to appear in training? It would be helpful to show figures of training/text losses w.r.t. number of epochs or iterations.
>
> We would like to emphasize that when we say “early into training” we are speaking specifically to the relative height along the loss surface at which the generalization gap occurs, i.e. that the generalization gap occurs at a training loss that is several orders of magnitude higher than the eventual training loss.
>
> While this generalization gap does also occur early into training from the perspective of epochs as well (i.e. nearly all generalization gaps we observe arise within the first 10% of training epochs taken) we focus on comparing trajectories across their training loss because multi-task models typically optimize individual task losses slower than single-task models due to task conflict. As a result, an epoch-wise comparison of generalization might show a generalization gap between multi-task and single-task models, but it would be unclear if this gap were due to how multi-task training impacted the speed at which the task loss was minimized or if the gap were due to factors detached from the training loss. By instead comparing across the training loss, we are able to show that these generalization gaps occur due to factors detached from the training loss.

---

> > ### Author Response · Authors · 2024-11-21
> > **Response to Reviewer yNT4 continued**
> >
> > > In Section 4, the paper observes that no factor is capable of explaining negative and positive transfer in any MTL setting we study. What is the correlation between the factors to negative transfers? Can we also measure the factors at the checkpoint at each epoch to study the correlation? How about considering various subset combinations of tasks (such as in GLUE), and how do the factors correlate with the transfer results (it would be better to report correlation scores)?
> >
> > > I also have similar questions about the results in Figure 4. It seems that there is some correlation between the factors and the transfer results. It remains unclear how strong/weak the correlation is. For example, can one fit a linear or polynomial model using the values of factors to predict the transfer results? How significant would that prediction be? It would be better to show similar figures like Figure 5 or 6
> >
> > In Sections 4 and 5, our goal is to determine whether there is some factor that can explain why, in a fixed multi-task setting, some tasks experience positive transfer and others experience negative transfer. We did not report correlation coefficients here because it was not clear to us that the values of a factor, and the delta of that factor between single-task and multi-task models, were comparable across different tasks. For example, should observing a change in sharpness by +1 and a change in accuracy by -5% in task A tell us anything about what accuracy to expect from a change in sharpness by +2 for task B? Without making some strong assumptions, we would argue that we cannot necessarily infer that the decrease in accuracy will be worse than 5%; however, we could still infer that the change will be negative if we assume that sharpness is correlated with generalization in task B.
> >
> > For that reason, in Sections 4 and 5, we focus on comparing positive and negative deltas, rather than computing a correlation between the actual values of each factor across different tasks. This is because, for a factor to explain why one task benefits while another suffers, it would be sufficient to show that this factor improves over STL for tasks that experience positive transfer and is worse than STL for tasks which experience negative transfer. We make this comparison using colors in Table 4, and shaded regions in Figure 4. However, we do not observe such a connection in practice, i.e. none of these factors can explain the trade-offs in task transfer for STL vs. MTL, or UMTG vs. SMTOs.
> >
> > Regarding comparing correlations across various subsets, if we understand the question correctly, this may be the experiment that we perform in Section 6, albeit not for the GLUE setting. Here, computing correlation coefficients is clearly valid because we are computing correlations over factors across the same task trained in different settings (i.e. with different subsets of auxiliary tasks) rather than computing correlations over factors across different tasks.
> >
> > > It would be better to provide a section to describe the practical implications of the observations: such as how can we choose MTL optimization algorithms given the evaluated factors in the early phase of training (sharpness, Fisher, gradients)?
> >
> > Thank you for this suggestion! We have modified the end of section 6 to make our practical implications more clear.
> >
> > > There are prior works on predicting negative transfers in multitask learning, which should be discussed in the paper:
> > [1] Vu, T., Wang, T., Munkhdalai, T., Sordoni, A., Trischler, A., Mattarella-Micke, A., ... & Iyyer, M. (2020). Exploring and Predicting Transferability across NLP Tasks. In Proceedings of the 2020 Conference on Empirical Methods in Natural Language Processing (EMNLP)
> > [2] Li, D., Nguyen, H., & Zhang, H. R. (2022) Identification of Negative Transfers in Multitask Learning Using Surrogate Models. Transactions on Machine Learning Research.
> >
> > Thank you for pointing out these relevant works, we have added a discussion of them to the draft in section D.

---

> ### Comment · Reviewer_yNT4 · 2024-11-24
> **Official comment from Reviewer yNT4**
>
> Thanks so much for the author's detailed responses! They have addressed most of my concerns. Thus, I would like to change the **Claims And Evidence** to Yes.
>
> I think it would be better to provide more quantitative descriptions in the papers, such as "the first 10% of training epochs early into training" mentioned in the responses, and the correlation scores between various factors.
>
> Moreover, there are developments in estimating task affinity scores (for characterizing task transfers). It would be better to discuss these works in the paper:
> - Which Tasks Should Be Learned Together in Multi-task Learning? ICML 2020
> - Boosting multitask learning on graphs through higher-order task affinities. KDD 2023
> - Scalable Multitask Learning Using Gradient-based Estimation of Task Affinity. KDD 2024

---

### Review · Reviewer_oXuy · 2024-10-14

**Summary Of Contributions:**

The paper offers a thorough analysis of the generalization challenges in multi-task learning, exploring its connections to training losses, optimization trajectories, positive/negative transfer, and gradient similarity. Through extensive experiments and ablation studies, the authors conclude that current multi-task optimization strategies do not significantly differ in their effects on optimization trajectories and its relationships with generalisation are still unclear. This finding may reinforce concerns about the limited generalization of various multi-task optimization methods, as previously highlighted in other studies.

**Audience:**

Yes

**Claims And Evidence:**

Yes

**Requested Changes:**

Fix references. Address the weaknesses listed above.

**Strengths And Weaknesses:**

This paper is well-written and presents its findings with strong experimental support and a clear takeaway, which I really appreciate. Below, I offer some minor comments in the order they appear in the paper:

1. The paper covers multiple aspects of MTL generalization, including training dynamics and gradient conflicts, yet the current title only emphasizes optimization trajectories. To better reflect the paper's scope, I would suggest adopting a more general title that reflects the full range of topics discussed.

2. The references are inconsistent in formatting—some use abbreviated conference names while others do not; and quite a few references completely miss the conference/journal names. Additionally, the 7-class Cityscapes dataset citation might be incorrect; the dataset was constructed in MTAN, not MGDA.

3. The paper concludes that “multi-task transfer is a property of how gradient conflict impacts the early phase of learning.” However, why is this limited to the early phase? While Fig. 1 shows that gradient conflict can be identified early on, this does not necessarily imply that conflicts are **only exisiting** in the early phase.

4.  The paper convincingly argues that modified optimization trajectories introduced by SMTOs do not fully explain generalization behaviour in MTL, which I agree with. However, this observation may extend beyond MTL and apply to single-task learning scenarios as well, such as those involving data augmentation, loss regularization, or neural architecture variations. This actually suggests a broader phenomenon in deep learning, where generalization in high-dimensional parameter spaces remains poorly understood.

5. At the end of Section 5, the paper concludes that mitigating gradient conflict does not necessarily improve generalization. Does this imply that MTL or auxiliary learning optimization strategies which were designed with a meta-objective directly optimized for generalization performance are inherently better? For instance, approaches like AuxiLearn (ICLR 2021), Auto-Lambda (TMLR 2022), and Task Weighting in Meta-learning with Trajectory Optimization (TMLR, 2023) could be relevant here. Looking foward to hearomg authors' comments on this.

---

> ### Author Response · Authors · 2024-11-21
> **Response to Reviewer oXuy**
>
> We would like to thank the reviewer for their time and efforts toward improving our paper!
>
> Below we attempt to address your listed concerns:
>
> > The paper covers multiple aspects of MTL generalization, including training dynamics and gradient conflicts, yet the current title only emphasizes optimization trajectories. To better reflect the paper's scope, I would suggest adopting a more general title that reflects the full range of topics discussed.
>
> Thank you for this suggestion - our hope in using the term “Optimization Trajectories” is that it encapsulates many properties of training dynamics (including sharpness, FIM, gradient covariance, and gradient conflict) rather than just e.g. the training loss. We attempted to make this clear in the abstract, by tying the phrase “factors of the optimization trajectory” to prior work which has studied phenomena such as implicit regularization. However, we are open to alternative phrases that more accurately capture our scope.
>
> >The references are inconsistent in formatting—some use abbreviated conference names while others do not; and quite a few references completely miss the conference/journal names. Additionally, the 7-class Cityscapes dataset citation might be incorrect; the dataset was constructed in MTAN, not MGDA.
>
> Thank you for noting this, we have addressed this in the revised draft!
>
> > The paper concludes that “multi-task transfer is a property of how gradient conflict impacts the early phase of learning.” However, why is this limited to the early phase? While Fig. 1 shows that gradient conflict can be identified early on, this does not necessarily imply that conflicts are only exisiting in the early phase.
>
> Thank you for this question. It is certainly the case that gradient conflict can, and does, exist well after the initial phase of training. However, based on our findings in Sec 3. concerning generalization gaps, we argue that the mechanisms that drive transfer are impacted early into training, and we are therefore interested in how gradient conflict might impact those mechanisms in the initial phase of training.
>
> > The paper convincingly argues that modified optimization trajectories introduced by SMTOs do not fully explain generalization behaviour in MTL, which I agree with. However, this observation may extend beyond MTL and apply to single-task learning scenarios as well, such as those involving data augmentation, loss regularization, or neural architecture variations. This actually suggests a broader phenomenon in deep learning, where generalization in high-dimensional parameter spaces remains poorly understood.
>
> We agree that even in single-task settings, generalization in deep learning remains poorly understood. Our goal in this paper (to understand whether there is a connection between optimization and generalization in multi-task learning) was motivated by the prevalence of multi-task optimizers proposed in recent work around multi-task learning coupled with the observation from many that these methods rarely work well in practice. To that end, our results demonstrate why SMTOs fail to consistently improve the performance of MTL models.
>
> We believe our work has more general ramifications for the understanding of generalization beyond multi-task learning. We show that several factors previously believed to be connected to the large-batch generalization gap (particularly FIM Explosion and Covariance) do not explain multi-task generalization gaps and therefore cannot be the sole drivers of generalization gaps in deep learning.
>
> > At the end of Section 5, the paper concludes that mitigating gradient conflict does not necessarily improve generalization. Does this imply that MTL or auxiliary learning optimization strategies which were designed with a meta-objective directly optimized for generalization performance are inherently better? For instance, approaches like AuxiLearn (ICLR 2021), Auto-Lambda (TMLR 2022), and Task Weighting in Meta-learning with Trajectory Optimization (TMLR, 2023) could be relevant here. Looking foward to hearomg authors' comments on this.
>
> Indeed, for goals such as auxiliary task selection, we feel it is likely the case that methods which use signals from e.g. validation data to learn task relationships are inherently more principled than approaches which use gradient conflict of the training objectives, because the relationship between gradient conflict and transfer is still poorly understood.

---

> > ### Comment · Reviewer_oXuy · 2024-11-30
> >
> > Thanks for your detailed response. I have no further questions and am happy to keep my original rating.

---

### Review · Reviewer_KAUz · 2024-11-11

**Summary Of Contributions:**

The paper investigates why certain multi-task settings enhance performance over single-task learning. The authors revisit existing analyses, particularly from an optimization perspective, and conclude that none of these approaches directly predict the generalizability of models trained with multi-task learning. However, they empirically observe that the positive or negative impacts of multi-task learning are established in the early stages of training.

**Audience:**

Yes

**Broader Impact Concerns:**

There are no broader impact concerns for this paper.

**Claims And Evidence:**

No

**Requested Changes:**

**[Experiments]**
As mentioned in Weakness, it would be nice to emphasize CelabA, Cityscapes, and GLUE results and please elaborate more on the results.


**[Writing]**
* The phrase “Beyond the zero’th order” is used too frequently (5 times), and the word “beyond” appears 8 times; consider rephrasing to avoid repetition.
* Change “zero’th” to “zeroth.”
* In Figure 5, add “and p-value” to the background color setting.
* In Appendix B, correct “FashionMNISt” to “FashionMNIST” and “lenet” to “LeNet.”
* In Figure 4, the legend locations vary by row, and the notation for the Gradient Method differs in the second row. Please standardize these, for example, by having a single legend per row. Additionally, the straight lines from (0,0) to each data point don’t add value and could be removed.
* Figures 5 and 6 could be combined into a single figure (or arranged as subfigures), or at least placed on the same page. Currently, readers must switch between pages to follow the authors’ claims effectively.

**[Question]**
* In Figure 2, what do the green and red bars represent, and what does each dot signify?
* Why does the last column in Figure 11 have an extremely large error bar?

**Strengths And Weaknesses:**

**[Strength]**
1. The paper addresses an essential aspect of multi-task learning: identifying which combinations of tasks create synergy and how to predict these interactions, although a clear conclusion on this remains open.
2. Table 1 effectively illustrates the relationship between generalization performance changes and various optimization factors.
3. The authors conduct a wide range of experiments.


**[Weakness]**

1. The proposed FashionMTL dataset requires a more detailed explanation, and it may not be well-suited for analyzing multi-task learning. First, the authors mention splitting FashionMNIST into two parts (Fashion1 and Fashion2) while maintaining class balance. If each split includes all classes and has the same distribution, it’s unclear whether this setup can genuinely be considered “multi-task” — it seems more like a simplified example. Additionally, the test set should be clarified: in NoisyFashion, are the test labels also shuffled?

2. FashionMTL, MNISTS, and CIFAR-100 are not standard multi-task datasets, which typically involve predicting different components (e.g., depth, segmentation, and attributes) from a single input. Instead, these datasets comprise multiple single-task settings that share the same backbone network and use ground-truth task labels to assign inputs to each classifier. Using datasets like Multi-Digit MNIST [2] or Taskonomy [3] would be more suitable for this context.

3. While “generalization” in multi-task learning is often compared to single-task learning (e.g., [1]), the term “generalization gap” requires further clarification, especially regarding the comparison at the same training loss level. For example, achieving the same training loss does not necessarily guarantee equivalent performance.
4. The description of gradient similarity (Eq. 6 and Eq. 1) needs elaboration. For instance, are gradients from all layers concatenated, or is a specific layer, such as the second-to-last, used?
5. There is no explanation provided for why each task shows a different trend in Figure 6. Additionally, the choice of auxiliary tasks (1, 4, 9, and 19) should be further explained, including the rationale behind selecting these particular numbers.


[1] Hu, Ziniu, et al. "Improving multi-task generalization via regularizing spurious correlation." Advances in Neural Information Processing Systems 35 (2022): 11450-11466.

[2] Sun, S.-H. Multi-digit mnist for few-shot learning, 2019. URL https://github.com/ shaohua0116/MultiDigitMNIST.
[3] Zamir, Amir R., et al. "Taskonomy: Disentangling task transfer learning." Proceedings of the IEEE conference on computer vision and pattern recognition. 2018.

---

> ### Author Response · Authors · 2024-11-21
> **Response to Reviewer KAUz**
>
> We would like to thank the reviewer for their review, and their time spent improving our submission.
>
> Below, we respond to the reviewers listed weaknesses:
> > The proposed FashionMTL dataset requires a more detailed explanation, and it may not be well-suited for analyzing multi-task learning [...]
>
> Thank you for the feedback; we have included more detail in the revised paper. Our objective with the FashionMTL setting is to demonstrate a synthetic multi-task setting in which we have a target-task (Fashion1) and 2 auxiliary task settings: the first is an idealized auxiliary task that results in positive transfer; the second is instead where the auxiliary task contains no helpful information and results in negative transfer. For the first setting, we treat the second half of FashionMNIST (Fashion2) as an auxiliary task will always yield positive transfer because it is more data from the same distribution, i.e. there is almost no task-conflict; for the second setting, we turn Fashion2 into NoisyFashion by shuffling the labels, which results in negative transfer. Under this, we have a synthetic, but idealized, setting in which to study negative and positive transfer.
>
> We do not evaluate our model on the test sets of Fashion2 or NoisyFashion, as our focus in this setting is strictly on how the target-task (Fashion1) is affected.
>
> >FashionMTL, MNISTS, and CIFAR-100 are not standard multi-task datasets, which typically involve predicting different components (e.g., depth, segmentation, and attributes) from a single input...
>
> While it is true that many multi-task settings involve predicting different components from a single input, we argue that such datasets (perhaps more adequately named multi-objective datasets) represent a subset of all multi-task datasets. CIFAR-100 was used as a test setting in the original PCGrad paper [1]; GLUE, which we study in this paper, was explicitly proposed as a multi-task benchmark [2]; additionally, recent studies of multi-task optimizers have included experiments on e.g. multilingual settings [3, 4]. Thus, while it may be useful to differentiate between settings in which we have multiple labels for a single input versus different labels for distinct inputs, we argue that both settings constitute a multi-task learning problem where transfer and task conflict are of interest.
>
> Moreover, we emphasize that we do consider multi-objective datasets in Cityscapes and CelebA experiments. By considering settings where we predict components of a single input and where we predict different components over distinct inputs, we show that optimization fails to explain multi-task transfer across a much broader spectrum of multi-task learning.
>
> >While “generalization” in multi-task learning is often compared to single-task learning (e.g., [1]), the term “generalization gap” requires further clarification [...]
>
> We adopt the term generalization gap from literature on the large batch generalization gap [5], in which it is common to compare task generalization by training loss for different solutions (e.g. Fig 3 in [5]) to demonstrate that the difference in generalization between large-batch and small-batch solutions cannot be explained by how well large-batch and small-batch trajectories minimize the training loss. In section 3, we demonstrate that this observation is also true when comparing multi-task and single-task models. We have attempted to clarify this by making the connection to prior literature more clear.
>
> >The description of gradient similarity (Eq. 6 and Eq. 1) needs elaboration. For instance, are gradients from all layers concatenated, or is a specific layer, such as the second-to-last, used?
>
> Thank you for noting this, we have made this clearer in the current draft. We consider gradients for all layers concatenated together, e.g. the gradient of the entire model.
>
> [1] Yu et al., 2020; Gradient Surgery for Multi-Task Learning
>
> [2] Wang et al., 2018; GLUE: A Multi-Task Benchmark and Analysis Platform for Natural Language Understanding
>
> [3] Wang et al., 2020; Gradient Vaccine: Investigating and Improving Multi-task Optimization in Massively Multilingual Models
>
> [4] Xin et al., 2022; Do Current Multi-Task Optimization Methods in Deep Learning Even Help?
>
> [5] Keskar et al., 2017; On Large-Batch Training for Deep Learning: Generalization Gap and Sharp Minima

---

> > ### Author Response · Authors · 2024-11-21
> > **Response to Reviewer KAUz continued**
> >
> > >There is no explanation provided for why each task shows a different trend in Figure 6. Additionally, the choice of auxiliary tasks (1, 4, 9, and 19) should be further explained, including the rationale behind selecting these particular numbers.
> >
> > That each task shows a different trend in generalization versus task conflict in Figure 6, despite each factor of optimization showing strong correlation with task conflict in Figure 5, is in fact the point: task conflict cannot explain multi-task transfer in Figure 6. In other words, our explanation for the different trends in Figure 6 is that transfer is not strongly correlated with task conflict.
> >
> > The choice of the numbers of auxiliary tasks was a decision on our part that covers (roughly) doubling the number of tasks at each interval until the maximum number of auxiliary tasks are used (i.e. we go from 2 to 5 to 10 to 20 total tasks in each case). The actual auxiliary tasks that are selected are sampled randomly and we use the same set of auxiliary tasks for each target-task across all 3 target-tasks. We have clarified both of these points in the current draft.
> >
> > >In Figure 2, what do the green and red bars represent, and what does each dot signify?
> >
> > The bars represent 2 standard deviations of the mean across random seeds for each training loss bin. Each dot signifies the training loss and generalization performance of a single checkpoint (evaluation of a single random seed at a given epoch).
> >
> > >Why does the last column in Figure 11 have an extremely large error bar?
> >
> > Some values in this plot are 0, or at least so close to 0 that the precision we use (8 decimal points) does not capture them. As a result, the standard deviation appears very large in log-scale.
> >
> > Finally, we have incorporated the reviewers requested changes into the revised version of the paper, and we thank them again for their many helpful suggestions.

---

> > ### Comment · Reviewer_KAUz · 2024-11-21
> >
> > I appreciate the authors for their response.
> >
> > Regarding FashionMTL, I believe that it is good to mention that both tasks use the entire labels in case of confusion.
> >
> > Although I still have concerns about whether FashionMTL is a good dataset to present the authors' contribution, considering the remaining parts and authors' clarification, I do not have further concerns.

---

### Author Response · Authors · 2024-11-21
**General Response**

We would like to thank all of the reviewers for their time and feedback.

In addition to responding to the reviewers individually, we have attached a revised version of the paper with the reviewers requested changes. Significant changes to the text are shown in red, while minor details (such as typos) are fixed without color.

Thank you again for your efforts to improve our work.

---

### Decision · Action_Editor_12Ep · 2024-12-09

**Recommendation:** Accept as is

**Comment:**

This paper investigates when and how multi-task learning might improve generalization, by looking at learning trajectories during optimization.  The reviewers all found that the claims were well supported by the comprehensive empirical evidence.  They also felt that the paper and the presented takeaways would be of interest to the TMLR community.  Overall, the reviews were very positive, and the reviewers applauded the comprehensiveness of the empirical analysis and found the presented discussion and conclusions interesting and convincing.  The authors did well to address the reviewers' concerns during the discussion and the reviewers appreciated the clarifications and proposed revisions.  The paper presents a clear empirical study that addresses a highly relevant hypothesis in ML.  I found it interesting that the authors found a generalization gap between single task and multi-task early on the training process, emphasizing the importance of early training dynamics.  There was a bit of a negative result in terms of whether the various metrics studied were indicative of positive task transfer (I.e. none of them were), but this is well supported by the experiments and also an interesting takeaway.

**Audience:**

The notion that training on multiple tasks to improve generalization is certainly of interest to the ML community, and this paper seems to do a good job of shedding some light on what happens during optimization that might explain this improved generalization.

**Claims And Evidence:**

The reviewers all found that the claims were well justified by the evidence provided.  In particular, they all emphasized that the study was quite comprehensive and provided clear takeaways.